*Resource*

EMBO
Molecular Medicine

# IFN-gene signatures in B cells following influenza A and B virus infection and influenza vaccination

Wuji Zhang[1,2], E Kaitlynn Allen [3], Shihan Li[4], Ilariya Tarasova [5], Rubaiyea Farrukee[1],
Lukasz Kedzierski [1], Brad Gilbertson [1], Hayley A McQuilten [1], Jennifer R Habel[1], Lilith F Allen [1],
Steven Rockman[1,6], Sarah L Londrigan[1], Stephen J Kent [1], Adam K Wheatley [1], Jason A Trubiano[7,8,9,10],
Tom C Kotsimbos[11,12], Allen C Cheng[13,14], Jan Schroeder[4], Jeremy Chase Crawford [15], Paul G Thomas [16],
Katherine Kedzierska [1,17]✉ & Thi H O Nguyen [1,17]✉

## Abstract

**Influenza viruses continue to cause a substantial global disease burden. Despite influenza vaccination, some individuals succumb to life-threatening influenza or death. Yet our understanding of immune features elicited by vaccination and influenza A and B virus (IAV, IBV) infection is limited. To define molecular signatures of influenza-specific B-cells, we performed scRNA-sequencing of influenza-specific B-cells in vaccinees and hospitalized IAV/IBV-infected patients using HA-probes. We observed increased interferon-stimulated gene signatures (*IF44L*, *IFITM1* and *XAF1*), in total B-cells from IBV-patients, but not at 1-month following patients' recovery or in IAV-patients or vaccinees. Phenotypic differentiation and isotype class-switching of HA-specific B-cells were observed following vaccination, with clonal sharing between memory and atypical B-cell phenotypes. In-vitro influenza virus infection experiments showed IBVs having higher infectivity of human PBMCs, including B-cells, and reduced B-cell proliferation compared to IAV, potentially associated with antiproliferative effect of IFITM1. We provide key insights into B-cell immunity towards IBV and IAV infections and vaccination, which will inform rational vaccine design and therapeutic strategies aimed at eliciting robust HA-specific B-cell responses, while minimizing adverse effects caused by natural infection.**

**Keywords** Influenza virus infection; Influenza vaccination; influenza-specific B cells; atypical B cells; BCR
**Subject Categories** Immunology; Microbiology, Virology & Host Pathogen Interaction

## Introduction

Influenza A and influenza B viruses (IAV, IBV) continue to cause a substantial global disease burden, with IBV associated with more severe clinical outcomes in children (Zaraket et al, 2021). Humoral immune responses provide key protection against influenza virus infections, primarily by antibodies targeting the haemagglutinin (HA) protein secreted by differentiated B cells such as antibody-secreting cells (ASCs) and plasmablasts (Chen et al, 2018). Differentiation and class-switching of B cells is aided by CXCR5[+] T follicular helper (Tfh) cells in secondary lymphoid organs (Schmitt et al, 2014). Previous studies demonstrated that ASCs and activated (PD-1[+] and/or ICOS[+]) circulating CXCR5[+]CXCR3[+]CCR6[-] Tfh1 cells positively correlated with anti-body responses following vaccination with inactivated influenza vaccines (IIV) and influenza virus infection (Bentebibel et al, 2013; Koutsakos et al, 2018; Nguyen et al, 2021; Nguyen et al, 2024). Similarly, IgD[-] class-switched memory HA-specific B cells increased following influenza virus infection and IIV vaccination (Koutsakos et al, 2018; Nguyen et al, 2021).

It is unknown, however, how influenza-specific B cells elicited by inactivated influenza virus vaccination and infection differ at the molecular level. Previous reports investigating patients with severe and mild influenza revealed more interferon (IFN)-related

[1]Department of Microbiology and Immunology, University of Melbourne, at the Peter Doherty Institute for Infection and Immunity, Parkville, VIC 3010, Australia. [2]HKU-Pasteur Research Pole, School of Public Health, LKS Faculty of Medicine, The University of Hong Kong, Hong Kong SAR, China. [3]Department of Immunology, St. Jude Children's Research Hospital, Memphis, TN 38105, USA. [4]Computational Sciences Initiative, Department of Immunology and Microbiology, University of Melbourne, at the Peter Doherty Institute for Infection and Immunity, Melbourne, VIC 3000, Australia. [5]Bioinformatics Division, Walter and Eliza Hall Institute of Medical Research, Parkville, VIC 3052, Australia. [6]Vaccine Product Development, CSL Seqirus Ltd, Parkville, VIC 3052, Australia. [7]Department of Infectious Diseases, University of Melbourne, at the Peter Doherty Institute for Infection and Immunity, Melbourne, VIC 3000, Australia. [8]Department of Infectious Diseases, Centre for Antibiotic Allergy and Research, Austin Health, Heidelberg, VIC 3084, Australia. [9]Department of Infectious Diseases, Peter MacCallum Cancer Centre, Melbourne, VIC 3000, Australia. [10]National Centre for Infections in Cancer, Peter McCallum Cancer Centre, Melbourne 3000 VIC, Australia. [11]Department of Respiratory Medicine, The Alfred Hospital, Melbourne 3004 VIC, Australia. [12]Department of Medicine, Central Clinical School, The Alfred Hospital, Monash University, Melbourne 3004 VIC, Australia. [13]School of Public Health and Preventive Medicine, Monash University, Clayton 3800 VIC, Australia. [14]Monash Infectious Diseases, Monash Health and School of Clinical Sciences, Monash University, Clayton 3168 VIC, Australia. [15]Department of Host-Microbe Interactions, St. Jude Children's Research Hospital, Memphis 38105 TN, USA. [16]Immunology and Vaccine Development Program, Vaccine and Infectious Disease Division, Fred Hutchinson Cancer Center, Seattle 98109 WA, USA. [17]These authors contributed equally: Katherine Kedzierska, Thi H O Nguyen. ✉E-mail: kkedz@unimelb.edu.au; thonguyen@unimelb.edu.au

transcripts in less severe influenza virus infections, whereas those requiring mechanical ventilation showed more inflammatory transcripts (Dunning et al, 2018). Within immune cell subsets (e.g. B cells and plasmablasts), increased IFN-γ and IFN-α response pathways were associated with severe and fatal influenza and COVID-19 (Mudd et al, 2020). The role of IFN signalling has been also described during IIV vaccination. Individuals with higher influenza microneutralization antibody titres following IIV exhibited upregulation of IFN-stimulated genes (ISGs), including *STAT1* and *IRF7* in peripheral blood (Gonçalves et al, 2019). Within a subset of activated B cells with high *FCRL5* and *ITGAX* expression, vaccine-responsive cells were also enriched for IFN signalling genes, including *IRF7*, *IFITM1* and other ISGs (Wang et al, 2023a). However, the specific involvement of the IFN pathway in B cells following infection versus vaccination remains unclear.

To investigate transcriptomic profiles during influenza virus infection and following influenza vaccination, as well as between IAV infection versus IBV infection, we performed single-cell RNA sequencing (scRNAseq), including hash-tagging of samples, barcode-labelled HA-probes and CITE-Seq antibodies, on total B cells and HA-specific B cells from hospitalised IAV- and IBV-infected individuals during acute illness and 30 days post-infection, and individuals vaccinated with IIV. The usage of BCR segments in HA-specific B cells was also investigated. Our analyses revealed that *ITGAX*-expressing atypical B cell responses were elicited following IIV vaccination. Influenza virus infection, especially IBV, led to strong IFN responses during acute infection in B cells, including HA-specific B cells. In vitro influenza virus infections revealed that IBV exhibits pronounced tropism and cytotoxicity towards immune cells. Additionally, IBV infection resulted in lower B cell proliferation compared to IAV, which may be attributable to the antiproliferative effects of IFITM1. These findings have important implications to inform future vaccine and treatment strategies against influenza virus infections. Specifically, targeting the transcriptomic profiles of HA-specific B cells may help to elicit more robust B cell responses.

# Results

## Cohort study design

To define single-cell transcriptomic profiles of influenza-specific B cell responses during influenza virus infection and vaccination, we analysed IAV- and IBV-specific B cells from hospitalised influenza virus-infected patient "Dissection of influenza-specific immunity" (DISI) cohort (Nguyen et al, 2021) and participants from the influenza vaccination Fluvax cohort (Koutsakos et al, 2018). Acute samples from hospitalised influenza-infected patients (IAV1, IAV2, IBV1 and IBV2 patients, aged 21–63 years, all female) were collected at 5-18 days post disease onset (between 2 and 4 days post hospital admission), while follow-up samples were collected following hospital discharge at 27–51 days post disease onset (Fig. 1A). Healthy influenza vaccinees (VAX1 and VAX2 participants, aged 25–27 years, both female) were blood sampled on day 0 pre-vaccination and on d7 and d28 post-vaccination (Fig. 1A). Patients were all admitted to the respiratory/general ward, with hospital stays ranging between 2 and 7 days, and all survived to 30 days post-admission. IAV2 required non-invasive oxygen

support, while all except IAV1 received oseltamivir treatment. IAV1 and IBV2 were considered highly susceptible to severe influenza disease due to underlying chronic respiratory disease and immunosuppression. IBV2 also had cardiac disease and chronic renal disease (Table EV1). Patients' total cytokine levels were moderate (1500–2000 pg/mL) in comparison to healthy vaccinees, except VAX2 had increased levels of IL-8 (5000 pg/mL) on d28 (Appendix Fig. S1A). Haemagglutination inhibition (HAI) titres increased from acute infection to follow-up and d0 to d28 post-vaccination, reaching 40 and above except for IBV1 (Appendix Fig. S1B). Antibody responses towards other subgroup strains were also boosted in patients at follow-up (Appendix Fig. S1C). In line with antibody responses, all participants had prototypical antibody-axis immune responses following infection or vaccination, with antibody-secreting cells (range 5.24–81.5 cells/mL) and PD-1$^+$ICOS$^+$-activated Tfh type-1 cells (4.47–28.18 cells/mL) highest at acute/d7 timepoints, and influenza-specific memory B cells (108.04–418.76 cells/mL) highest at follow-up/d28 timepoints (Appendix Fig. S1D).

## Phenotypically distinct B cell clusters within influenza virus infection and vaccination

HA-specific B cells and CXCR5$^+$ Tfh cells from each participant and timepoint were isolated for single-cell RNA sequencing using hash-tagging oligos (HTO), barcoded HA-probes and barcoded CITE-Seq antibodies (Fig. 1B,C; Appendix Fig. S2A). A total of 32,502 sorted cells were sequenced with highly valid HTO barcode rates (>82.9%) and sequencing saturation rates (>87.1%) (Appendix Fig. S2B). Following quality control, 28,028 cells (86.2%) were retained for further analyses, including 6171 B cells and 21,857 Tfh cells (Appendix Fig. S2C–E). B cell clusters with high *CD19* expression were re-clustered into 13 unsupervised clusters, with cluster 3 divided into 3 clusters (clusters 3_0, 3_1 and 3_2) due to variable gene expression and clusters 1 and 10 combined as cluster 1 due to minimal differences in gene expression (Fig. 1D). As commonly used B cell phenotyping markers, CITE-Seq CD21 and CD27 levels were able to differentiate between naïve-like (CD21$^{hi}$CD27$^{lo}$, clusters 0, 1, 6, 7 and 8), memory-like (CD21$^{lo/hi}$CD27$^{hi}$, clusters 2, 3_0, 4, 5, 9 and 12) and atypical-like B cell clusters (CD21$^{lo}$CD27$^{lo}$, clusters 3_1 and 3_2) (Fig. 1E).

Annotation of each B cell cluster based on their differentially expressed genes (DEGs) further confirmed their B cell phenotype as well as their activation status (Fig. 1F; Appendix Figs. S3 and S4). For example, clusters 3_1 and 3_2 representing CITE-Seq CD21$^{lo}$CD27$^{lo}$ atypical B cells showed higher expression of *FCRL5* and *FCRL3* (Fig. 1G; Appendix Fig. S4), both of which are highly expressed in atypical or CD21$^{lo}$ activated memory B cells based on influenza vaccination and other infection studies in humans (Holla et al, 2021; Nellore et al, 2023; Sutton et al, 2021). Cluster 3_1 also expressed *ITGAX* (Fig. 1G), which encodes for CD11c and has been used to further define atypical B cells as CD11c$^+$CD21$^{lo}$CD27$^{lo}$ in influenza vaccination studies (Sutton et al, 2021). Clusters 0, 1, 6, 7 and 8 all expressed one or more genes highly expressed in naïve B cells (Wen et al, 2020), including *FCER2*, *TCL1A* and *IL4R*. Memory B cell clusters 2, 3_0, 4, 5, 9 and 12 were further defined as activating (cluster 3_0: *CD53*; 4: *TRAF4*) and/or proliferating B cells (cluster 2: *CD1C*; 5: *SLC3A2*) according to their specific gene sets (Fig. 1F; Appendix Fig. S3-S4).

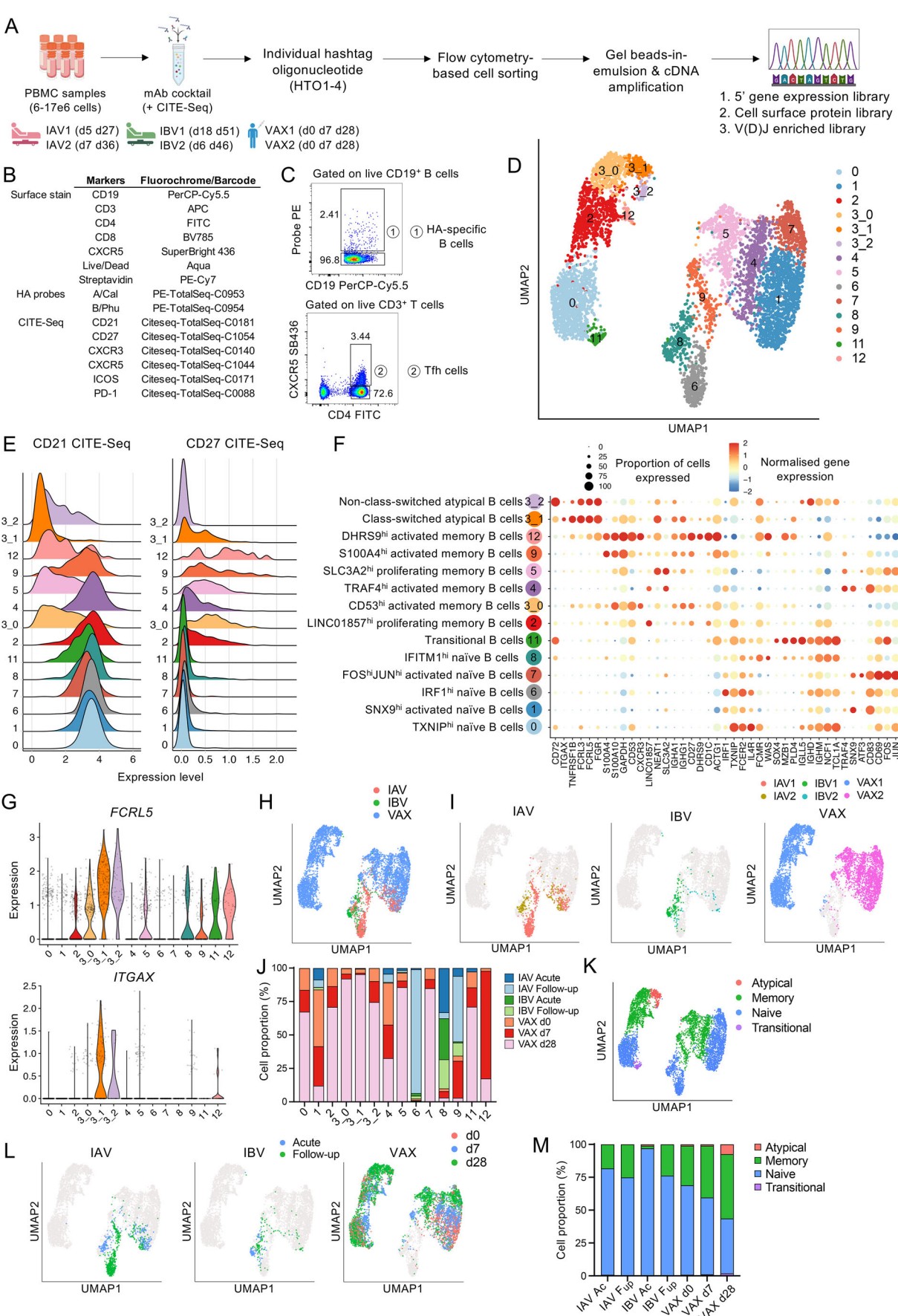

**Figure 1.   10X scRNAseq and cluster annotation.**

(**A**) Workflow and sample timepoints; (**B**) Antibody panel; and (**C**) gating strategy of FACS cell sorting and 10X scRNAseq ($n = 2$ biological replicates per IAV, IBV and VAX groups). (**D**) Unsupervised clustering of sorted B cells. (**E**) Histogram plots of CD21 and CD27 CITE-Seq expression levels. (**F**) Manual cluster annotation and selected top DEGs per B cell cluster. (**G**) Violin plots of *FCRL5* and *ITGAX* per cluster ($n = 6171$ cells). (**H, I**) UMAP showing B cell projection grouped by (**H**) infection/vaccination status and (**I**) individual. (**J**) Stacked bar graph of distribution of infection/vaccination timepoints per cell cluster. (**K, L**) UMAP showing B cell projection grouped by (**K**) annotated phenotypes and (**L**) timepoints following infection or vaccination. (**M**) Stacked bar graph of cell phenotype per condition.

To determine whether infection/vaccination status and time-points contributed to B cell clustering, sample groups were mapped onto B cell clusters, revealing two main regions of vaccination, separated based on the two vaccine participants (VAX1 and VAX2) rather than timepoints, and one main region of infection (Fig. 1H–J). VAX1 occupied naïve cluster 0, transitional cluster 11, memory clusters 2/3_0/12 and atypical clusters 3_1/3_2 on the left of the UMAP, while VAX2 occupied naïve clusters 1/7 and memory clusters 4/5 on the right. The IAV acute timepoint was mainly in naïve cluster 8, while the IAV follow-up was mainly in clusters 6/9. IBV infection at acute and follow-up was mainly in cluster 8 (Fig. 1H,J). Genes associated with B cell differentiation also contributed to the B cell clustering with a gradual progression from naïve to memory to atypical B cells within the vaccination regions (Fig. 1K,L). Memory B cell subsets increased gradually following vaccination, with the emergence of atypical B cells at d28 post-vaccination (Fig. 1M). Overall, we found phenotypically distinct B cell clusters within IAV/IBV infection and following vaccination.

## Differentially expressed B cell activation genes following influenza vaccination

Human influenza vaccination studies have previously highlighted upregulation of genes responding to BCR and IFN signalling in B cells post-IIV (Wang et al, 2023a; Wang et al, 2023b). To determine which genes were enriched in our study, we compared gene profiles of B cells at d0, d7 and d28 timepoints following IIV, based on a twofold change threshold in gene expression ($log_2FC \geq 1$) (Fig. 2A–C). *CXCR4* expression, which encodes for the chemokine receptor CXCR4 expressed on B cells during B-cell development and trafficking to lymphoid tissues and inflammatory sites, was higher at d0 and d28 when compared to d7 (Fig. 2A,B). Upregulation of *CXCR4* in B cells has been observed following IIV, especially in a subset of activated B cells (Horns et al, 2020). The decrease in *CXCR4* at d7 post-vaccination is possibly due to B cells migrating into the tissues, as upregulation of *Cd69* and *Cxcr4* in memory B cells has been described in the lungs of mice following IAV A/Puerto Rico/8/34 infection (Mathew et al, 2021). As observed in previous studies with various stimulus including anti-IgM (Fowler et al, 2013), B cells at d28 post-vaccination had higher *FOS* expression, indicating activation of B cells (Fig. 2B). Pathway analyses also revealed "Cytokine Signalling in Immune system" pathway was involved in all three timepoint comparisons, indicating importance of cytokine signalling during influenza vaccination (Fig. 2D).

To investigate whether differences in gene expression were driven by their B cell phenotypes, we analysed gene profiles between different vaccination timepoints for naïve B cells (i.e. clusters 0, 1, 6, 7 and 8) and memory B cells (i.e. clusters 2, 3_0, 4, 5, 9 and 12) (Fig. 2E,F). While downregulation of *CXCR4* on d7

compared to d0 was observed in both naïve and memory B cells, key activation genes (*FOS*, *JUN*, *CD69*) were only upregulated in naïve B cells but not memory B cells on d28 compared to d0 or d7, suggesting de novo B cell responses elicited following IIV from the naïve B cell compartment (Turner et al, 2020). Our data highlight *CXCR4*, *FOS*, *JUN* and *CD69* as the key genes associated with B cell activation that are differentially expressed during influenza vaccination.

## Vaccine-induced clonal expansion of influenza-specific B cells

To investigate single-cell transcriptomes of influenza-specific B cells during IIV or influenza virus infection, we utilised fluorescently labelled barcoded recombinant HA (rHA) probes specific for A/H1N1/California/7/2009 (A/Cal) and B/Yamagata/Phuket/3073/2013 (B/Phu) influenza virus strains. Both strains were represented in the 2015 and 2016 IIV and in the hospitalised-infected patients. Manual gating was applied to single A/Cal-specific and B/Phu-specific B cells (Fig. 3A; Appendix Fig. S5A). B cells outside of the manual gates but sharing BCR sequences to cells within the A/Cal and B/Phu gates (named A/Cal clonal and B/Phu clonal) represented 4.5% and 2.9% of total A/Cal-specific (33/729) and B/Phu-specific (25/871) B cells, respectively (Appendix Fig. S5A). The majority of clonally-expanded B cells were influenza-specific (86.6%) and derived from the vaccination samples (96.1%). Clonal expansion was also observed from the infection samples (3.9%) (Appendix Fig. S5A), although this group represented 5.9% of total influenza-specific B cells (95/1600 cells) and was not included in the transcriptomic analyses.

Shared BCR clonotypes were found across different vaccination timepoints within the same donors, but not across donors (Fig. 3B; Appendix Fig. S5B). Clonal size increased from 1–2 cells on d0 to up to 7 A/Cal+ and 33 B/Phu+ clonal cells on d28 post-vaccination, with most clonotype sharing between d7 and d28. We have previously shown that influenza-specific B cells differ in their phenotype and isotype in response to IIV and influenza virus infection using flow cytometry (Nguyen et al, 2021). UMAP analyses showed majority of A/Cal clonal and B/Phu clonal B cells (i.e. outside of manual gate but shared BCRs) clustered together with other influenza-specific B cells from the manual gates, thus revealing similar transcriptomic profiles between those "clonal" B cells and influenza-specific B cells (Fig. 3C). Interestingly, influenza-specific B cells were mostly located in clusters 3_0, 3_1, 3_2 and 5 at d28 post-vaccination (Fig. 3C), which represented more differentiated atypical and memory gene profiles (Fig. 3D). Those cells found in clusters 3_1 and 3_2 highly express *FCRL5* (Fig. 1F) and is consistent with other human IIV studies demonstrating the importance of *FCRL5*+ or FcRL5+T-bet+ cells within influenza-specific B cells (Burton et al, 2022; Nellore et al, 2023). One study showed that FACS-sorted FcRL5+ influenza-

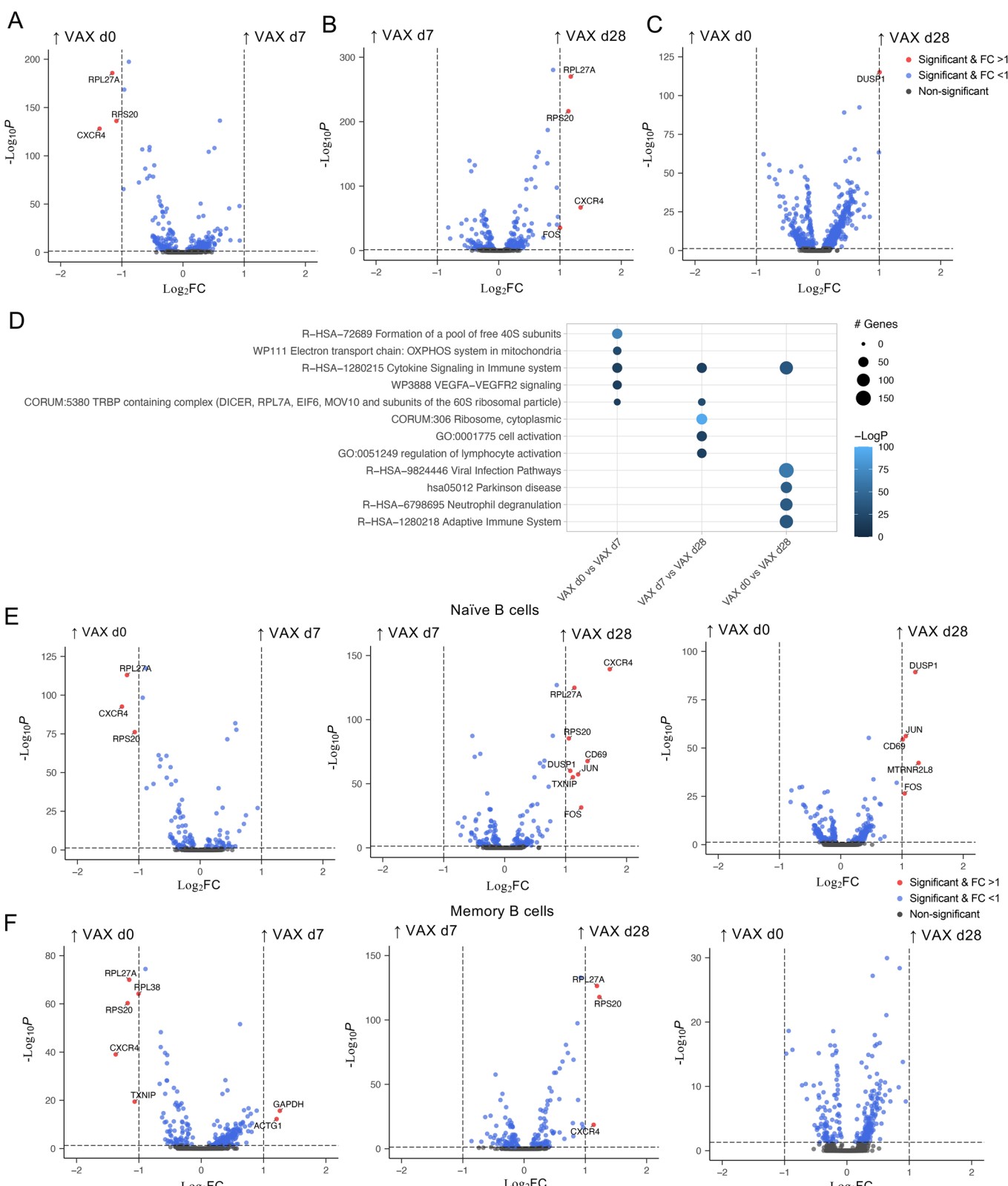

**Figure 2. Higher B cell migration marker at VAX d7, while higher activation at VAX d28.**

(A–C) Volcano plots showing fold-changes of DEGs of total B cells for (A) VAX d0 vs VAX d7, (B) VAX d7 vs VAX d28 and (C) VAX d0 vs VAX d28 ($n = 2$ biological replicates for VAX timepoints; $n_{VAX\,d0} = 1078$ cells, $n_{VAX\,d7} = 987$ cells, $n_{VAXd28} = 3007$ cells). (D) Bubble plot of the top five enriched terms for each comparison across different vaccination timepoints. (E, F) Volcano plots showing fold-changes of DEGs of (E) naïve B cells (clusters 0, 1, 6, 7, 8; $n_{VAX\,d0} = 734$ cells, $n_{VAX\,d7} = 577$ cells, $n_{VAXd28} = 1256$ cells) and (F) memory B cells (clusters 2, 3_0, 4, 5, 9, 12; $n_{VAX\,d0} = 324$ cells, $n_{VAX\,d7} = 389$ cells, $n_{VAXd28} = 1474$ cells) for VAX d0 vs VAX d7, VAX d7 vs VAX d28 and VAX d0 vs VAX d28. Statistical significance for DEGs were calculated using the Wilcoxon rank-sum test with $P$ values adjusted with a Bonferroni correction using all genes in the dataset.

specific B cells exhibit elevated gene expression involved in BCR signalling and mitochondrial activity, further validating their activated and effector status in response to IIV (Nellore et al, 2023).

Concordant with the differentiated phenotypes, BCR heavy chain constant segments revealed decreasing proportion of influenza-specific B cells expressing less differentiated *IGHD* and *IGHM* gene segments but increasing proportions of class-switched *IGHG1* and *IGHA1* gene segments by d28 post-vaccination compared to d0 and d7 (Fig. 3E). Similar projection was observed for clonally-expanded influenza-specific B cells, where the bulk of clonally-expanded A/California and B/Phuket HA-specific B cells were detected on d28 and mapped to clusters 3_0, 3_1, 3_2 and 5 (Fig. 3F), representing memory or atypical phenotypes (Fig. 3G) and also enriched for *IGHG1* and *IGHA1* gene segments (Fig. 3H).

We next determined which genes were commonly enriched between A/California and B/Phuket HA-specific B cells following IIV, in comparison to total B cells from earlier analyses (Fig. 3I). Like the total B cell analyses, which identified four key DEGs, few genes were differentially expressed between vaccination timepoints for A/California and B/Phuket HA-specific B cells (Fig. 3I), and even less so between A/California and B/Phuket HA-specific B cells at one timepoint, such as d7 post-vaccination (Appendix Fig. S5C). *CXCR4* was indeed down-regulated on d7 post-vaccination for total B cells and A/California and B/Phuket HA-specific B cells (Fig. 3I left panel). However, upregulation of *CXCR4* on d28 post-vaccination was only observed in total B cells (Fig. 3I middle panel), suggesting that perhaps influenza-specific B cells are residing longer within the tissues following vaccination, as observed in the mouse infection model (Mathew et al, 2021). Genes associated with BCL-6 suppression, *TXNIP* and *ITM2B* (Baron et al, 2015; Shao et al, 2010), were down-regulated on d7 post-vaccination compared to d0 in B/Phuket HA-specific B cells (Fig. 3I), coinciding with upregulation of *CD1C*, which is involved in BCR-induced activation (Allan et al, 2011). *CD1C* and *GAPDH* was also enriched in d7 B/Phuket HA-specific B cells compared to d7 A/California HA-specific B cells, whereas the latter population was enriched for many ribosomal genes, albeit at log$_2$FC <1 (Appendix Fig. S5C).

Overall, we observed gradual increases in A/California and B/Phuket influenza-specific B cells and accumulation of class-switched BCR heavy chain genes following IIV vaccination. These influenza-specific B cells were highly clonally expanded by the d28 population, representing a more differentiated memory and atypical transcriptional profile.

## BCR gene segment sharing between memory and atypical influenza-specific B cells

We next investigated the BCR repertoire of influenza-specific B cell responses elicited following vaccination. High levels of clonally-expanded A/California (38%) and B/Phuket HA-specific B cells (53%) were found in our IIV dataset, which were most abundant on d28 post-vaccination and often shared between d7 and d28 within donors and HA-probe (Fig. 4A). This is in line with our previous IIV study where >57% of influenza-specific B cells were clonally expanded using PCR methods (Koutsakos et al, 2018). BCR repertoire was then assessed based on their BCR heavy variable (*IGHV*) gene segment pairing with either lambda variable (*IGLV*) or kappa variable (*IGKV*) chains.

While overall BCR segment usage was diverse, heavy chains *IGHV3-23* (light yellow) and *IGHV3-30* (yellow) were frequently detected in A/California HA-specific B cells from both vaccinees at all timepoints (8–22%, except that *IGHV3-30* (yellow) was not detected for VAX1 at d0), with two clonotypes detected across multiple timepoints in VAX2 (Fig. 4B–D, Appendix Fig. S6A–C; Dataset EV1). For VAX1, various *IGHV* gene segments were mainly paired with *IGLV2-8* (cyan) or *IGKV3-15* (light teal), including expanded clonotypes (*IGHV1-46/IGLV2-8*, *IGHV3-7/IGKV3-15* and *IGHV3-23/IGKV3-15*). Similarly, two light chain segments, *IGLV2-8* (cyan) and *IGKV3-15* (light teal), were detected in VAX2 with diverse heavy chains but were less clonally expanded than VAX1.

For B/Phuket HA-specific B cells, heavy chains *IGHV3-30* (yellow) and *IGHV3-30-3* (golden yellow) were observed in both donors at all timepoints, with the most seen at d28 with multiple *IGLV* pairings, representing 7.2% (57 out of 796) of our dataset (Dataset EV1). Similarly, we observed ~25% of B/Phuket HA-specific B cells having *IGHV3-30* or *IGHV3-30-3* in our previous IIV study (Koutsakos et al, 2018). On d28 post-vaccination, other variable gene pairings such as *IGHV5-10-1/IGKV3-20* and *IGHV3-15/IGLV6-57* in VAX1, and *IGHV3-7/IGKV3D-20* and *IGHV3-15/IGKV3-11* in VAX2 were also commonly observed (Fig. 4B–D; Appendix Fig. S6A–C).

We next investigated the BCR repertoire between B cell phenotypes and found that expanded clonotypes in VAX1 were shared between memory and atypical B cell subsets at 47.9% (23 out of 48) and 71.8% (28 out of 39) within A/California and B/Phuket HA-specific B cell populations, respectively (Fig. 4E; Appendix Fig. S6D). In contrast, these clonotypes were not shared with the naïve subset, except for 1 A/California HA-specific B cell that was found across naïve, memory and atypical phenotypes. Whereas less clonotype sharing was observed between phenotypes for VAX2, which had fewer atypical B cells for comparison. No clear biases were observed in *IGHV* or *IGLV/IGKV* gene usage for memory or atypical B cells. Similarly, for VAX1, the joining segments for heavy (*IGHJ*) and kappa/lambda (*IGKJ/IGLJ*) chains were commonly shared between memory and atypical B cell subsets for both A/California and B/Phuket HA-specific memory B cells, except for *IGHJ5* and *IGKJ5*,

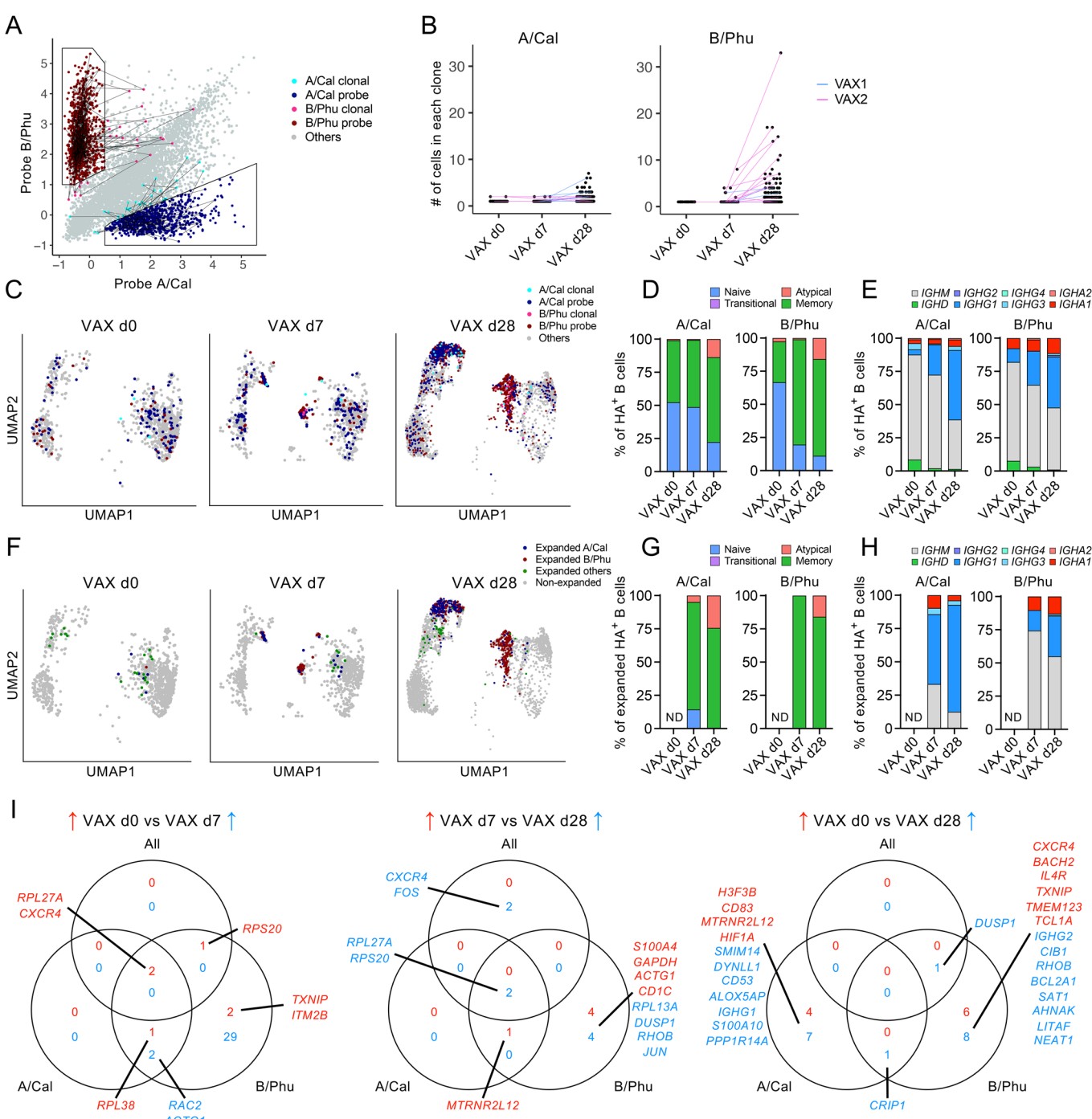

**Figure 3. Phenotype, isotype and DEGs of HA-specific B cells.**

(A) Gating of HA-specific B cells based on probe specificity and clonal B cells with shared BCR sequences. (B) Number of B cells in each B cell clone. Connected dots indicate HA-probe-specific clonal B cells with shared BCR sequences across timepoints. (C) UMAP projection, and (D) frequency of B cell phenotypes and (E) isotypes for HA-specific B cells. (F) UMAP projection, and (G) frequency of B cell phenotypes and (H) isotypes for clonally-expanded HA-specific B cells. Expanded B cell clones were defined as >1 B cell with shared BCR sequences. Phenotype and isotype analyses were only performed for samples with ≥10 B cells. (I) Venn diagrams showing shared and distinct significantly DEGs with log₂FC >1 between total B cells and HA-specific B cells for each comparison. Genes were listed when the number of genes was fewer than 10. The 29 DEGs specific for B/Phuket HA-specific B cells between VAX d0 and VAX d7 were *GAPDH, ATP5F1E, SH3BGRL3, COX6B1, PFN1, CLIC1, ACTB, TKT, PYCARD, ARPC1B, ACP5, ATP5MC3, CIB1, WAS, MAP4K1, CRIP1, PLAC8, PPP1CA, IFI30, ARPC2, FLNA, CD27, VIM, S100A4, CD1C, TPI1, TMEM156, UCP2, COTL1.* (A–I) n = 2 biological replicates for VAX timepoints.

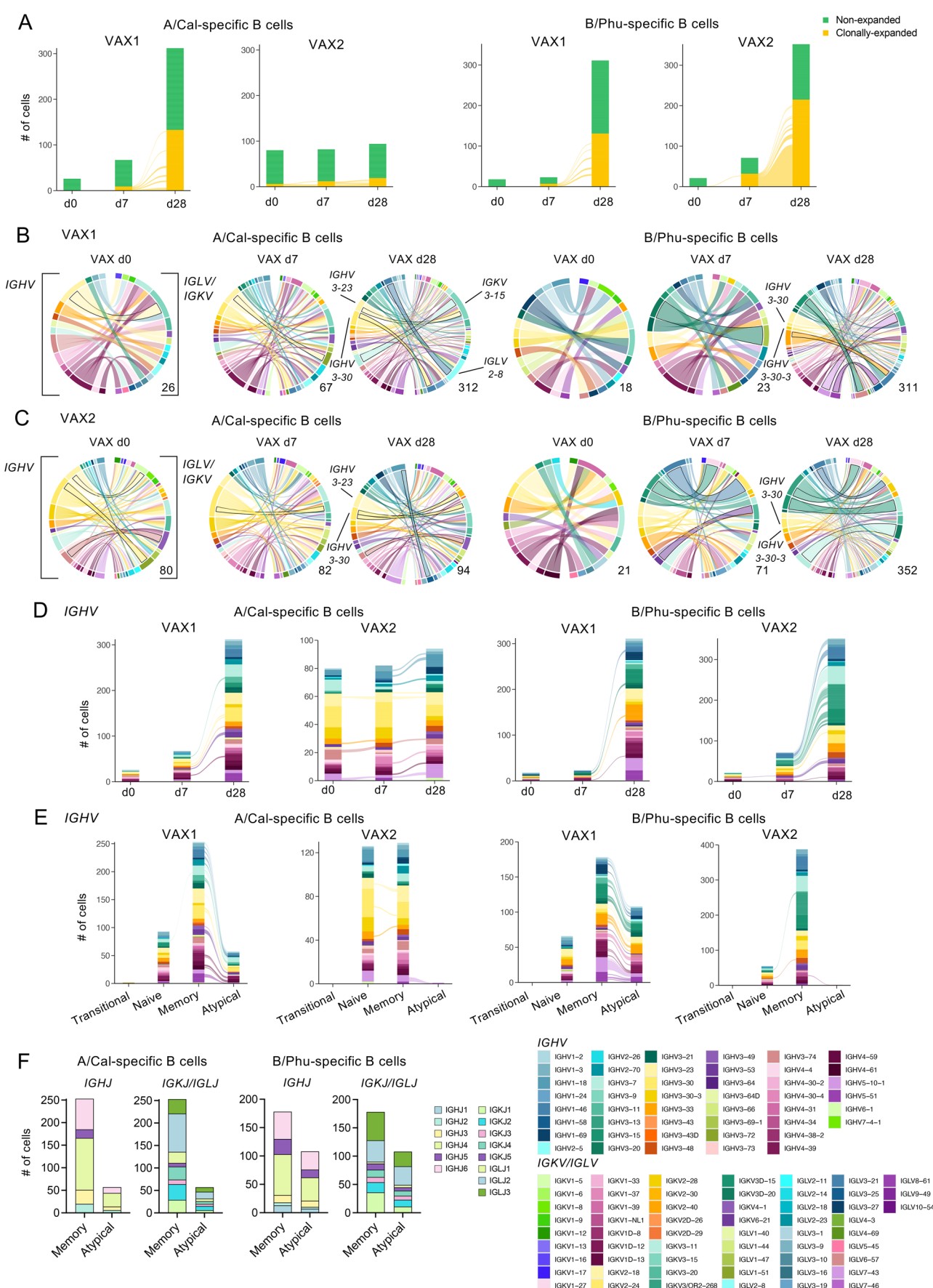

Figure 4. Clonality and BCR gene segment usages in HA-specific B cells.

(A) Alluvial plots of clonally-expanded and non-expanded HA-specific B cells. (B, C) Circos plots of *IGHV* and *IGLV/IGKV* gene segment pairings of all HA-specific B cells for (B) VAX1 and (C) VAX2. Outlines represent the most frequent segment pairings at each timepoint. (D, E) Alluvial plots of *IGHV* gene segment usage for all HA-specific B cells (D) across timepoints and (E) B cell phenotypes. (F) *IGHJ* and *IGKJ/IGLJ* usage for HA-specific B cells for VAX1.

which were not detected in A/California HA-specific atypical B cells (Fig. 4F).

Overall, clonal expansions of influenza-specific B cells peaked on d28 following IIV vaccination, and where atypical B cells were present, we observed a high degree of BCR repertoire sharing between memory and atypical B cells.

## Highly upregulated IFN signalling pathway during acute IBV infection

The transcriptomic differences in B cells between infections and vaccinations, as well as between IAV and IBV infections, are not yet fully understood. We first compared transcriptomic profiles between IAV infection and vaccination at the peak of the response (i.e. acute IAV and VAX d7) and at the follow-up/d28 timepoints within total, naïve and memory B cell subsets (Appendix Fig. S7). A small handful of DEGs were observed between IAV infection and vaccination at either timepoints, in which vaccination was enriched for *KLF2*, whereas infection was enriched for immune response genes *NFKBIA*, *NFKB1* and *IRF1*. KLF2 is a transcription factor that can act as an inhibitor for NK-κB signalling to reduce inflammation.

Next, we compared B cell transcriptomics between acute IAV and IBV timepoints in total and naïve B cell subsets. Memory B cell subsets (clusters 2, 3_0, 4, 5, 9, 12) could not be analysed due to lower cell numbers. Interestingly, several antiviral genes including *IFI44L*, *IFITM1*, *XAF1* and *IRF7* were highly elevated at acute IBV timepoint (Fig. 5A). This was supported by enriched term analysis of DEGs by Metascape (Zhou et al, 2019) revealing significant enrichment for the "Interferon Signalling" pathway as well as other important antiviral immune response pathways in B cells at acute IBV infection (Fig. 5B). Strikingly, antiviral genes such as *IFI44L*, *ISG15*, *IFITM1*, *IFITM2* and *IRF7* were expressed at higher levels at acute IBV infection in comparison to all other conditions, including acute IAV infection (Fig. 5C). Violin plots of *IFI44L*, *IFITM1* and *XAF1* confirmed that these genes were specifically upregulated during acute IBV infection in both IBV donors (Fig. 5D).

## Higher infectivity of B cells by IBV strains compared to IAVs

Given we observed enhanced activation of the IFN signalling pathway in B cells following IBV infections, we next asked whether there were any virus-mediated differences between IAV and IBV infection by utilising an in vitro influenza virus infection assay, which incubated healthy PBMCs with live IAV and IBV viruses at MOI of 1, 4 and 10 to assess infectivity rates in lymphocyte subsets and monocytes in a controlled in vitro setting (Fig. 6A–C; Appendix Fig. S8A). Interestingly, both B/Phuket and B/Malaysia strains showed higher infectivity of B cells based on intracellular NP-staining at 8 and 22 h post-infection compared to the A/California strain, which exhibited only minimal B cell infectivity.

Although both IBV strains consistently showed higher infectivity, A/California could infect CD19$^-$CD3$^-$CD56$^-$ cells (possibly monocytes), consistent with our previous report (Nguyen et al, 2021). Although the CD14 expression on CD19$^-$CD3$^-$CD56$^-$ cells was low following in vitro infection with live viruses, the frequency of monocyte-like cells, based on size and granularity, remained comparable across IAV, IBV and no virus conditions (Appendix Fig. S8B). Higher NP$^+$ frequency was also observed for various T cell subsets and NK cells following IBV infections (Fig. 6A–C), with NK cells showing IBV effects by 8 h post-infection.

To determine whether the inflammatory environment added to the effects of the infectivity between IAV and IBV, we additionally performed an in vitro influenza virus infection assay for 22 h at MOI = 4, but now using 10% heat-inactivated plasma obtained from patients hospitalised with either IAV, IBV, COVID-19 and RSV ($n = 20$; Fig. 7). Like using FCS in the cultures previously, IBV infection was still higher than IAV infection. However, compared to healthy plasma (obtained from 8 healthy participants) culturing conditions, plasma from IAV and IBV patients modestly increased the frequency of NP$^+$ B cells following IBV infections by 1.7–2.8-fold (Fig. 7A). A similar increase was observed in CD3$^-$CD19$^-$CD56$^-$ cells following A/California infection, as well as CD4$^+$ and CD8$^+$ T cells after IAV and IBV infections cultured with different plasma types (Fig. 7B–D). These findings suggest that, in addition to virus-specific factors mainly contributing to the IBV > IAV hierarchy, plasma mediators within acute hospitalised patient samples can also contribute to modest increases in NP$^+$ cell frequencies. Notably, that these plasma mediators are possibly shared across different respiratory viral infections, including IAV, IBV, COVID-19 and RSV.

## Dissecting the roles of the IFITM family following IAV and IBV infections

A previous study has shown that induced overexpression of IFITM1, IFITM2 and IFITM3 prior to infection restricts IAV infection in A549 cells, resulting in fewer infected cells and reduced viral titres (Meischel et al, 2021). To investigate whether these ISGs have direct antiviral effects during IBV infection in comparison to IAV infection, we evaluated the impact of inducible IFITM1 and three protein overexpression on IAV and IBV infections using A549 cell lines in the presence of doxycycline (Fig. 8A–D; Appendix Fig. S8C). Consistent with the previous study (Meischel et al, 2021), IFITM1 overexpression restricted the infectivity of A/California (H1N1 IAV) infection at 8 h with MOIs of 1 and 4, although infectivity of A549 cells was much lower (<10%) compared to IBV infections (~90%), a similar hierarchy pattern to the PBMC subsets, however IBV infectivity rates were not impacted by IFITM1 overexpression (Fig. 8E). IFITM3, on the other hand, significantly reduced the frequency of NP$^+$ A549 cells following both IAV and IBV infections (Fig. 8E), which was also observed earlier at 4 and 6 h of infection (Appendix Fig. S9).

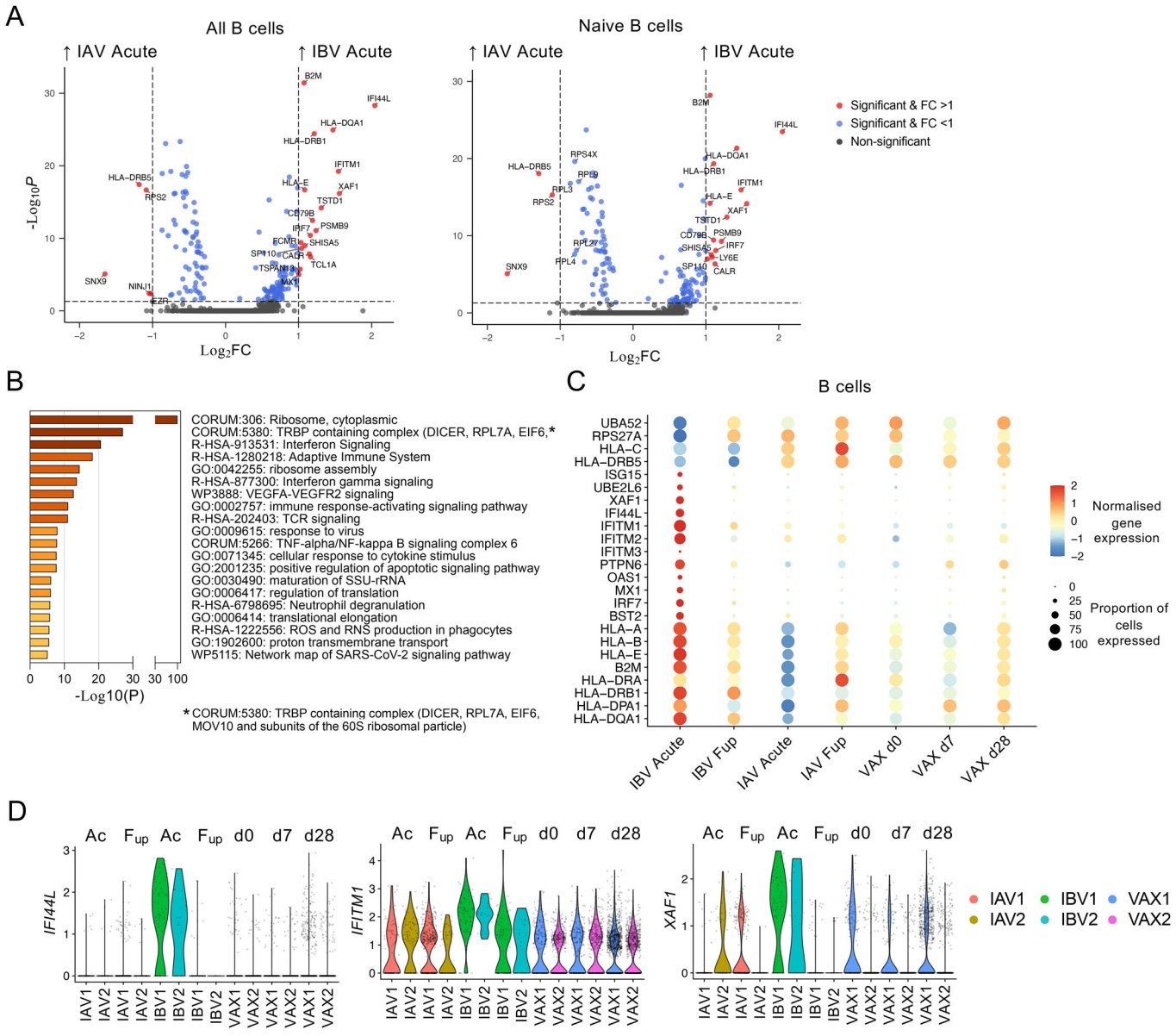

**Figure 5. DEGs between acute IAV and IBV timepoints.**

(A) Volcano plots showing fold-changes of DEGs for all ($n_{IAV\ Acute}$ = 252 cells, $n_{IBV\ Acute}$ = 101 cells) and naïve (clusters 0, 1, 6, 7, 8; $n_{IAV\ Acute}$ = 204 cells, $n_{IBV\ Acute}$ = 98 cells) B cells. (B) Enriched term analysis of all significant DEGs. (C) Bubble plot of DEG expression in B cells across different infection/vaccination timepoints. (D) Violin plots of *IFI44L*, *IFITM1* and *XAF1* expression ($n$ = 6171 cells). $n$ = 2 biological replicates per IAV, IBV and VAX groups. Statistical significance for DEGs were calculated using the Wilcoxon rank-sum test with $P$ values adjusted with a Bonferroni correction using all genes in the dataset.

Although IFITM1 was more highly expressed compared to IFITM3 from our single-cell 10x data on B cells, it did not have a clear effect on reducing IBV infections in the epithelial A549 cells, warranting further investigations of the impact of IFITM proteins on B cells.

Beyond direct interaction with viruses, anti-IFITM1 antibodies have been reported to inhibit proliferation of B cell lines and leukaemic B cells, suggesting a negative role for IFITM1 in pathways regulating cell growth (Bradbury et al, 1992; Evans et al, 1990). Therefore, we investigated whether elevated *IFITM1* expression in B cells following IBV infection, as observed in the 10X dataset, affects B cell proliferation. To promote B cell proliferation and differentiation, IL-21 and soluble CD40 ligand (sCD40L) were added to the culture (Auladell et al, 2019). Following infections, the number of live B cells was reduced after IBV infections compared to IAV at days 1, 4 and 6 post-infection, which was not boosted by presence of IL-21 and sCD40L (Fig. 8F,G; Appendix Fig. S8D). Rather, infection with A/California showed increases in live B cell numbers by day 6 of culture which was boosted by IL-21 and sCD40L, reflecting their ability to proliferate with a lower infectivity strain. B cell viability was

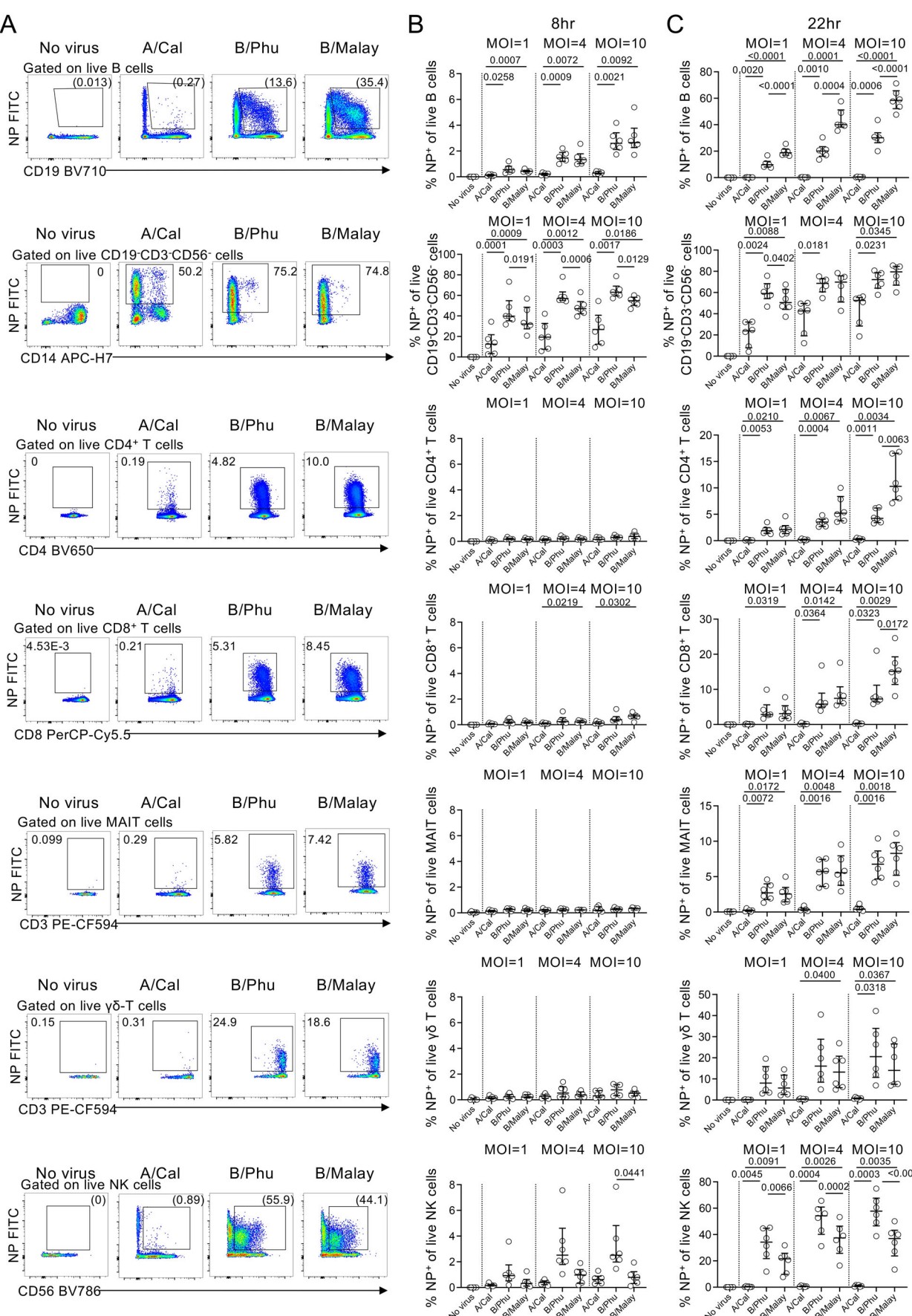

◀

**Figure 6.  Influenza virus infection assay of PBMCs with IAV and IBV.**

PBMCs were infected with A/California, B/Phuket, and B/Malaysia strains in vitro at MOI of 1, 4 and 10 for 8 h or 22 h ($n = 6$ biological replicates per group). Intracellular NP-staining was performed to investigate the proportion of cells that are NP positive. (**A**) Representative FACS plots of NP-staining on immune cell populations. (**B, C**) Frequencies of NP$^+$ cells at (**B**) 8 h and (**C**) 22 h post-infection. Bars indicate the median with IQR. Statistical significance was determined with RM one-way ANOVA with Geisser-Greenhouse correction followed by Sidak multiple comparisons. Exact $P$ values are shown ≥0.0001 and <0.05. Source data are available online for this figure.

significantly impacted by IBV infections on day 1 (58-74%) compared to A/California infection (76–85%), which caused a sharp decrease in viable cell numbers (Fig. 8H). Given the higher infectivity of IBVs observed in vitro, this suggested cytotoxic effects of IBVs on infected B cells. Further analysis showed that IAV infection led to greater numbers of B cells with more division cycles, while the proportion of dividing and non-dividing cells was similar between IAV and IBV infections (Fig. 8I,J). The addition of IL-21 and sCD40L also increased both the number and proportion of B cells undergoing divisions, including in the no virus control (Fig. 8I,J).

Overall, our human in vitro assays provide further insights into the transcriptomic differences observed in the B cell responses following IAV and IBV infections. IBV infections, at least in our model system, resulted in higher infectivity rates in B cells and lower B cell viability and proliferative capacity. Our findings may partially explain why we observed such strong upregulation of IFN signalling genes required for recovery from acute B/Phuket-infected patients, compared to our A/California-infected patients.

# Discussion

Although IIVs have been widely used for decades, their effectiveness varies across influenza strains and years (Belongia et al, 2016). Apart from strain mismatch, the heterogeneity of immune responses elicited by influenza virus infection and vaccination may be a factor leading to the suboptimal effectiveness, which highlights the importance of understanding the immunological mechanisms underlying these responses. To that end, we compared and contrasted transcriptomic profiles of B cells isolated after influenza virus infection and vaccination. Following IIV vaccination, we observed prominent clonal expansion of HA-specific B cells with differentiation and isotype switching. Antiviral gene expression, including *IFI44L*, *IFITM1* and *XAF1*, was predominantly upregulated during acute IBV infection. Subsequent in vitro experiments revealed distinct B cell responses to IAV and IBV infections.

In our scRNAseq analysis, we identified two B cell clusters at vaccination at d28 with transcriptomic features that resembled atypical B cells, including expression of *ITGAX*, *FGR*, *FCRL5* and *CD72* genes. This cell phenotype was predominantly in VAX1 and included A/California and B/Phuket HA-specific B cells. Interestingly, these atypical B cells were not found during and following IAV or IBV infection in our participants. Similar induction of this cell type following IIV has been previously reported. CD11c$^+$, CXCR3$^+$ and CD11c$^+$CXCR3$^+$ B cell subsets with A/California or B/Phuket HA-specificity were detected prior to IIV administration and increased in frequency post-vaccination (Sutton et al, 2021). Similarly, a separate adult cohort receiving IIV showed an increase in FCRL5$^+$ HA-specific atypical memory B cells (Burton et al, 2022). Pseudotime analysis based on transcriptomics has shown that, in malaria, atypical B cells differentiated from naïve B cells via

an IFN-γ-driven pathway, were distinct from the classical memory B cell lineage and exhibited minimal clonal sharing with memory B cells (Holla et al, 2021). A similar pseudotime trajectory was observed following influenza vaccination, suggesting that FCRL5$^+$ atypical memory B cells arise from naive B cells, potentially through an intermediate phenotype in the germinal centre (Burton et al, 2022). However, our BCR clonal expansion analysis revealed extensive clonal sharing between memory and atypical B cells, but not with naïve B cells. Transcriptomic analyses also showed atypical B cell clusters stemming from proliferating memory B cells and clustered near activated memory B cells. Therefore, our findings support a close relationship between memory and atypical B cells, while the lack of overlap with naïve B cells may reflect the rarity of HA-specific naïve B cell clones circulating in the blood.

Further investigation into the function of atypical B cells following influenza vaccination is essential to understand whether their responses might attenuate vaccine effectiveness. Atypical B cells can display an exhaustion-like phenotype. In malaria-exposed individuals, atypical B cells were less responsive to BCR signalling (Portugal et al, 2015). In contrast, co-culturing of atypical B cells with Tfh cells and Staphylococcal Enterotoxin B could induce upregulation of CD38 and secretion of IgG and IgM antibodies, suggesting these cells are still functional when receiving proper stimulation (Hopp et al, 2022). Moreover, CD11c$^+$CXCR3$^+$ HA-specific B cells exhibited features of both atypical lineage and recent activation (Sutton et al, 2021). FCRL5$^+$ atypical memory B cells demonstrated the highest rates of class-switch recombination and somatic hypermutation after IIV, suggesting involvement in germinal centre responses (Burton et al, 2022). While studies have shown the association between atypical B cells and germinal centre reaction, it is not the sole pathway to antibody production. In a patient with T-bet deficiency with low levels of atypical B cells, B cells were still able to differentiate into antibody-secreting plasmablasts (Yang et al, 2022). Overall, our findings support the idea that atypical B cells observed in our vaccinated donor likely represent a subset that has recently undergone activation in response to vaccination. However, these cells are not the sole pathway for B cell activation and antibody production, as illustrated by donor VAX2, who had minimal expansion of atypical B cells.

Our scRNAseq analyses revealed prominent IFN responses during acute IBV infection, compared to VAX d7 and IBV follow-up timepoints, as well as acute IAV infection in our patients. These responses were characterised by elevated expression of antiviral ISGs, including *IFI44L*, *ISG15*, *IFITM1* and *IRF7*. We then conducted in vitro influenza virus infection to further elucidate the potential role of ISGs. Interestingly, we observed a significantly higher frequency of NP$^+$ B cells, T cells and NK cells following IBV infection compared to IAV infection. Plasma from patients with influenza, COVID-19 or RSV, as compared to healthy plasma, modestly increased the infectivity of IAV and IBVs further, likely attributable to the highly inflammatory milieu present in the

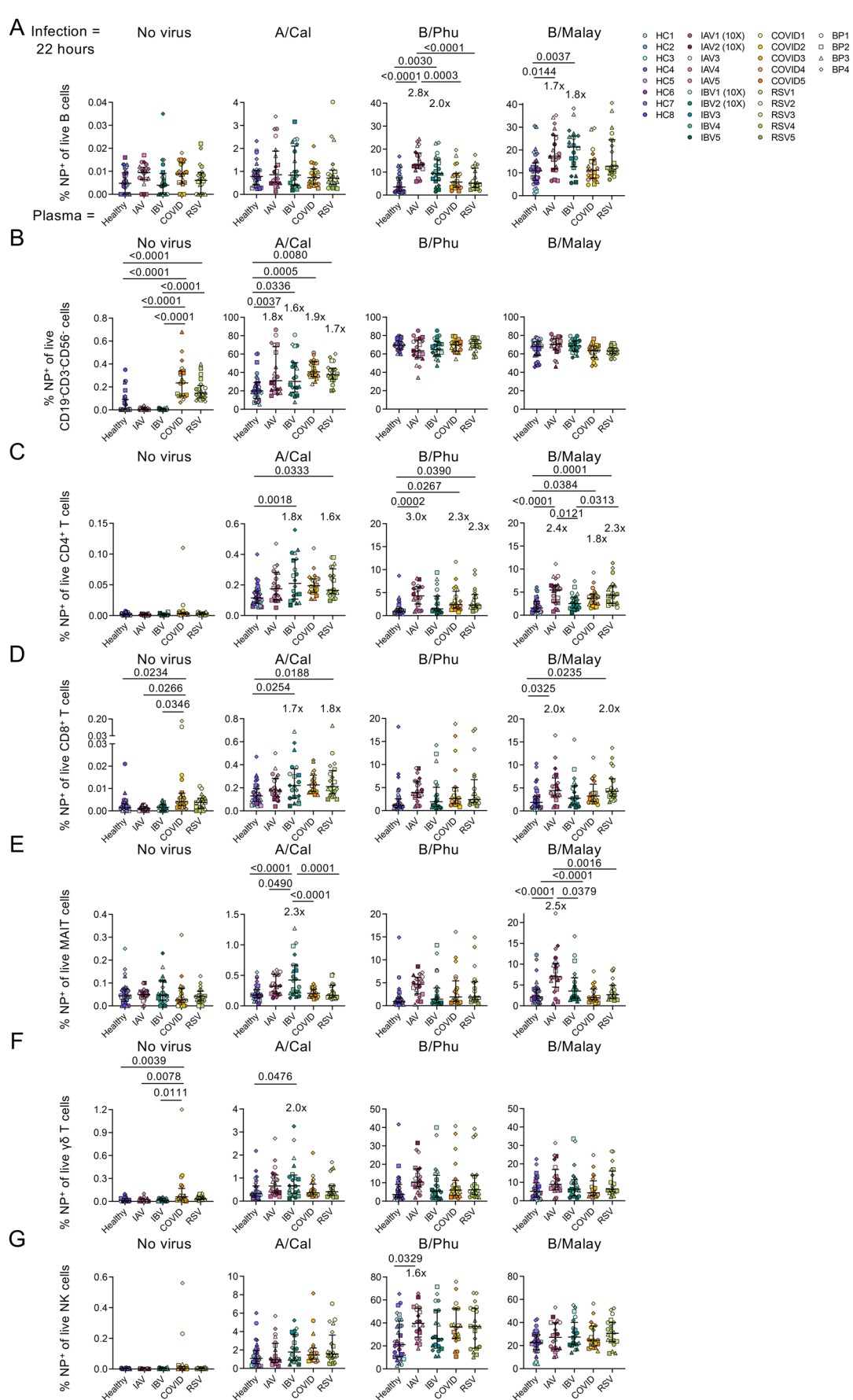

Figure 7. Influenza virus infection assay co-cultured with plasma from patients hospitalised with IAV, IBV, COVID-19 and RSV.

The assay was performed as described in Fig. 6 legend and Methods section at MOI = 4 for 22 h, with the exception that heat-inactivated patient plasma samples were added 1-h post-infection of influenza viruses. Experiments describe biological replicates from 4 buffy pack donors cultured with IAV (n = 5), IBV (n = 5), COVID-19 (n = 5), RSV (n = 5) and healthy plasma (n = 8), per virus condition. (A–G) The frequency of NP⁺ cells out of (A) B cells, (B) CD3⁻CD19⁻CD56⁻ cells, (C) CD4⁺ T cells, (D) CD8⁺ T cells, (E) MAIT cells, (F) γδ-T cells and (G) NK cells. Bars indicate the median with IQR. Statistical significance was determined with ordinary one-way ANOVA followed by Tukey's multiple comparisons. Exact P values are shown ≥0.0001 and <0.05. Source data are available online for this figure.

plasma. Influenza virus infection of adaptive immune cells has also been reported in previous studies. For instance, scRNAseq of lungs from IAV-infected mice indicated that 31% of B cells were infected, as measured by the presence of viral mRNA (Steuerman et al, 2018). However, expression of IFN-related genes was detected in both bystander and infected B cells, suggesting that upregulation of the IFN pathway does not require direct influenza virus infection (Steuerman et al, 2018). Therefore, although our dataset shows a strong IFN signature following IBV infection, we have no clear evidence whether these B cells were "more" infected in vivo, as we did not detect any viral transcript from our scRNAseq data. However, we also cannot rule this hypothesis out, as any viral transcript may have been undetectable due to the later sampling timepoints, which were between 5 and 18 days post disease onset. It has also been shown that IAV can infect HA-specific B cells via BCR interaction, rather than through sialic acid binding alone, resulting in impaired antibody production and B cell death (Dougan et al, 2013). Given the broader cellular tropism of IBV, it remains to be seen whether IBV can infect and deplete B cells in the respiratory tract and dampen mucosal B cell and antibody responses.

To further delineate the effects of ISG upregulation in B cells, we considered the established role of IFNs and ISGs as key modulators of antiviral immunity. IFNs and IFITMs, for example, can inhibit viral entry, replication and assembly either by directly targeting influenza viruses or by modulating host epithelial cells (Fong et al, 2022; Xu et al, 2022). Elevated IFN levels have been detected in the serum of both IAV- and IBV-infected patients within 6 days post disease onset (Bian et al, 2014; Galani et al, 2021). In accordance with our findings, IBV (B/Shangdong/7/1997) has been shown to induce peak expression of IFN-α, IFN-β and IFN-λ1 within 2–8 h post-infection in human monocyte-derived DCs and macrophages, whereas IAV (A/Beijing/353/1989) induced a delayed peak at 16-24 h (Mäkelä et al, 2015; Österlund et al, 2012). However, although B cells in our scRNAseq data exhibited a rapid upregulation of ISGs, our novel findings from the epithelial A549 cell lines suggest that IFITM1 expression may not confer the same level of antiviral protection against IBV as it does against IAV, warranting further investigation of these IFITM proteins in B cells.

Beyond the direct antiviral effects of the IFITM family, we investigated whether ISGs can modulate B cell function, as it was reported that in B cells, type I IFN can promote differentiation into plasma cells and enhance antibody secretion when co-culturing with influenza virus and T cells (Jego et al, 2003). Therefore, upregulation of IFN responses in B cells during acute IBV infection may impact humoral immune responses. We focused specifically on the IFITM protein family, which also plays an essential role in B cell responses. At steady state, IFITM1 and IFITM3 are expressed at low levels in B cells (Regino-Zamarripa et al, 2022); however, IFITM3 can be important for B cell expansion following BCR signalling (Lee et al, 2020). The specific role of IFITM1 in B cells remains less known. An early study in a leukaemia model found that IFITM1 may exert aggregation and antiproliferative effects on

B cells (Evans et al, 1990). Here, elevated IFITM1 expression was observed during acute IBV infection in our scRNAseq data, which may partially explain the reduced proliferation observed following in vitro IBV infection compared to IAV infection.

In conclusion, we observed differential transcriptomic profiles in B cells following IAV, IBV infections and influenza vaccination. HA-specific B cells underwent differentiation and isotype switching following vaccination and showed clonal sharing across memory and atypical B cell phenotypes. Excitingly, the IFN-mediated pathway was most prominently upregulated during acute IBV infection in our scRNAseq experiment. Further in vitro studies provided additional insights into IBV tropism and its cytopathogenic effects on different immune cells. While IFITM1 did not demonstrate direct inhibition of IBV infection, at least in the A549 cells, it may influence B cell proliferation. We acknowledge that our PBMC infection studies do not entirely reflect influenza virus infection of the respiratory mucosa; therefore, future in vitro studies harnessing lung organoid cells are warranted. Altogether, our findings provide much-needed knowledge for a detailed understanding of immune responses against IAV and IBV infections and vaccination responses. Our dataset provides proof-of-concept for exploring the link between HA-specific B cell clonality and transcriptomic signatures, as well as underscoring the need for further studies on the role of IFITM family members in B cell responses.

# Methods

**Reagents and tools table**

| Reagent/resource | Reference or source | Identifier or catalogue number | |
|---|---|---|---|
| **Experimental models** | | | |
| Peripheral blood samples (H. sapiens) | Nguyen et al, Nat Commun, 2021; Koutsakos et al, Sci Transl Med, 2018 | N/A | |
| A549 cells (H. sapiens) | ATCC | #CCL-185 | |
| A549 cells with DOX-inducible IFITM1, IFITM3, OVA (H. sapiens) | Meischel et al, J Virol, 2021 | N/A | |
| MDCK cells (C. familiaris) | ATCC | #CCL-34 | |
| **Recombinant DNA** | | | |
| **Antibodies** | **Reference or source** | **Identifier or catalogue number** | **Dilution** |
| **Cell sorting for single-cell RNA sequencing** | | | |
| Mouse anti-human CD4 FITC | BioLegend | Cat#300505 | 1:200 |

| Reagent/resource | Reference or source | Identifier or catalogue number | |
|---|---|---|---|
| Mouse anti-human CD19 PerCP-Cy5.5 | eBioscience | Cat#46-0198-42 | 1:200 |
| Mouse anti-human CD3 APC | BioLegend | Cat#300312 | 1:200 |
| Mouse anti-human CXCR5 SuperBright 436 | eBioscience | Cat#62-9185-42 | 1:200 |
| Mouse anti-CD8 human BV785 | BioLegend | Cat#301045 | 1:200 |
| Armenian Hamster anti-human ICOS Citeseq-TotalSeq-C0171 | BioLegend | Cat#313553 | 1:200 |
| Mouse anti-human PD-1 Citeseq-TotalSeq-C0088 | BioLegend | Cat#329963 | 1:200 |
| Mouse anti-human CXCR3 Citeseq-TotalSeq-C0140 | BioLegend | Cat#353747 | 1:200 |
| Mouse anti-human CXCR5 Citeseq-TotalSeq-C1044 | BioLegend | Cat#356939 | 1:200 |
| Mouse anti-human CD21 Citeseq-TotalSeq-C0181 | BioLegend | Cat#354923 | 1:200 |
| Mouse anti-human CD27 Citeseq-TotalSeq-C1054 | BioLegend | Cat#302853 | 1:200 |
| Mouse anti-human Hashtag 1 TotalSeq™-C0251 | BioLegend | Cat#394661 | 1:200 |
| Mouse anti-human Hashtag 2 TotalSeq™-C0252 | BioLegend | Cat#394663 | 1:200 |
| Mouse anti-human Hashtag 3 TotalSeq™-C0253 | BioLegend | Cat#394665 | 1:200 |
| Mouse anti-human Hashtag 4 TotalSeq™-C0254 | BioLegend | Cat#394667 | 1:200 |
| **Influenza virus infection assay** | | | |
| Mouse anti-human CD161 BV605 | BioLegend | Cat#339916 | 1:50 |
| Mouse anti- human CD4 BV650 | BD | Cat#563875 | 1:200 |
| Mouse anti- human CD19 BV711 | BD | Cat#563036 | 1:100 |
| Mouse anti- human CD56 BV785 | BD | Cat#564058 | 1:200 |
| Mouse anti- human CD14 APC-Cy7 | BD | Cat#560180 | 1:100 |
| Mouse anti- human CD8 PerCP-Cy5.5 | BD | Cat#565310 | 1:200 |
| Mouse anti- human TCR Vα7.2 PE | BioLegend | Cat#351706 | 1:400 |
| Mouse anti- human CD3 PE-CF594 | BD | Cat#562280 | 1:200 |

| Reagent/resource | Reference or source | Identifier or catalogue number | |
|---|---|---|---|
| Mouse anti- human TCR-γδ PE-Cy7 | BD | Cat#655410 | 1:50 |
| Mouse anti-IAV NP FITC | GeneTex | Cat#GTX36902-500UL | 1:200 |
| Mouse anti-IBV NP FITC | Invitrogen | Cat#MA1-7306 | 1:200 |
| **A549 cell infection assay** | | | |
| Rat anti-FLAG FITC | BioLegend | Cat#637308 | 1:200 |
| Mouse anti-IAV NP FITC | GeneTex | Cat#GTX36902-500UL | 1:200 |
| Mouse anti-IBV NP FITC | Invitrogen | Cat#MA1-7306 | 1:200 |
| **B-cell proliferation assay** | | | |
| Mouse anti- human CD71 BV650 | BioLegend | Cat#334116 | 1:200 |
| Mouse anti- human CD19 BV711 | BD | Cat#563036 | 1:100 |
| Mouse anti- human IgG BV785 | BD | Cat#564230 | 1:75 |
| Mouse anti- human CD27 APC | BD | Cat#558664 | 1:50 |
| Mouse anti- human CD20 AF700 | BD | Cat#560631 | 1:150 |
| Mouse anti- human CD14 APC-H7 | BD | Cat#560180 | 1:100 |
| Mouse anti- human CD69 PerCP-Cy5.5 | BioLegend | Cat#310926 | 1:50 |
| Mouse anti- human CD38 PE | BD | Cat#555460 | 1:50 |
| Mouse anti- human CD3 PE-CF594 | BD | Cat#562280 | 1:800 |
| Mouse anti- human IgD PE-Cy7 | BD | Cat#561314 | 1:500 |
| Mouse anti- human IgM BUV395 | BD | Cat#563903 | 1:150 |
| Mouse anti- human CD4 BUV496 | BD | Cat#612937 | 1:100 |
| Mouse anti- human CD21 BUV737 | BD | Cat#612788 | 1:100 |
| Mouse anti- human CD8 BUV805 | BD | Cat#564912 | 1:100 |
| Mouse anti-IAV NP FITC | GeneTex | Cat#GTX36902-500UL | 1:200 |
| Mouse anti-IBV NP FITC | Invitrogen | Cat#MA1-7306 | 1:200 |
| **Oligonucleotides and other sequence-based reagents** | **Reference or source** | **Identifier or catalogue number** | |
| PE-TotalSeq-C0953 | BioLegend | Cat#405265 | |
| PE-TotalSeq-C0954 | BioLegend | Cat#405267 | |

| Reagent/resource | Reference or source | Identifier or catalogue number | |
|---|---|---|---|
| **Chemicals, enzymes and other reagents** | **Reference or source** | **Identifier or catalogue number** | |
| Live/Dead Ghost Aqua | Tonbo | Cat#13-0870-T100 | |
| Live/Dead Aqua | Thermo Fisher | Cat#L34966 | |
| Live/Dead NIR | Thermo Fisher | Cat#L34976 | |
| Streptavidin-PE-Cy7 | eBioscience | Cat#25-4317-82 | |
| A/California rHA | Whittle et al, J Virol, 2014 | N/A | |
| B/Phuket rHA | Koutsakos et al, Sci Transl Med, 2018 | N/A | |
| CellTrace Violet | Thermo Fisher | Cat#C34557 | |
| Doxycycline | Sigma-Aldrich | Cat#D9891-5G | |
| Recombinant human IL-21 | PeproTech | Cat#200-21-10UG | |
| Soluble CD40 ligand | PeproTech | Cat#310-02-50UG | |
| **Software** | **Reference or source** | **Identifier or catalogue number** | |
| Vireo programme v0.2.3 | Huang et al, Genome Biol, 2019 | | |
| BD FACS Diva v8.0.1 | BD | https://www.bdbiosciences.com/en-au/products/software/instrument-software/bd-facsdiva-software | |
| FlowJo v10.10.0 | FlowJo | https://www.flowjo.com | |
| Prism v10.6.0 | GraphPad | https://www.graphpad.com | |
| R v4.3.1 | The Comprehensive R Archive Network | https://cran.r-project.org | |
| Seurat v4.3.0.1 | Hao et al, Cell, 2021 | | |
| SingleR v1.0 | Aran et al, Nat Immunol, 2019 | | |
| ggplot2 v3.4.3 | | https://cran.r-project.org/web/packages/ggplot2/index.html | |
| EnhancedVolcano v1.18.0 | | https://github.com/kevinblighe/EnhancedVolcano | |
| Metascape v3.5.20230623 | Zhou et al, Nat Commun, 2019 | | |
| Dandelion v0.3.3 | Syi et al, Nat Biotechnol, 2023 | | |
| Immcantation v0.3.2 | Gupta et al, Bioinformatics, 2015 | | |
| Circlize v0.4.16 | Gu et al, Bioinformatics, 2014 | | |
| ggVennDiagram v1.5.2 | Gao et al, Front Genet, 2021 | | |

| Reagent/resource | Reference or source | Identifier or catalogue number | |
|---|---|---|---|
| **Other** | **Reference or source** | **Identifier or catalogue number** | |
| 10x Genomics Chromium Single Cell 5' Library & Gel Bead Kit | 10x Genomics | 16 rxns PN-1000006 | |
| Cytofix/Cytoperm Kit | BD | Cat#554714 | |
| Vi-Cell XR Cell Viability Analyser | Beckman Coulter | | |
| SY3200 Cell Sorter | Sony | | |
| Chromium Controller | 10x Genomics | | |
| LSRII Fortessa | BD | | |

## Study participants

Hospitalised patients with influenza-like illness were recruited at the Alfred Hospital into the DISI cohort (Nguyen et al, 2021). Blood samples were collected upon hospital admission and at follow-up (~30 days). For scRNAseq, we used PCR-confirmed influenza A and influenza B patients recruited between May and October in 2017, apart from IAV1 who was recruited in September 2014 (Alfred Hospital Ethics Committee #280/14). The patients were infected with A/H1N1 (IAV1 and IAV2) or IBV (IBV1 and IBV2). All patients, apart from IAV1, received oseltamivir antiviral treatment, while IAV2 also received non-invasive oxygen support. IAV1 and IBV2 were both receiving immunosuppressants (Table EV1). For in vitro experiments, PBMCs were isolated from buffy coat samples obtained from Australian Red Cross Lifeblood under ethics 2015#8 (seven individuals, three males and three females, one unknown). For the plasma experiments, we used plasma from 20 patients hospitalised with IAV, IBV, RSV and COVID-19 ($n = 5$ for each infectious disease), recruited at Alfred Hospital (ethics number #280/14) and Austin Health (ethics numbers HREC/15/MonH/64, SSA/28204/Austin-2022), as well as eight healthy participants, recruited at the University of Melbourne (ethics number #13344).

Participants receiving IIV were recruited under the University of Melbourne Ethics number 1443389 as previously described (Koutsakos et al, 2018). Both vaccinees received the Fluvax IIV (bioCSL, VIC, Australia). VAX1 received the 2015 trivalent IIV, and VAX2 received the 2016 quadrivalent IIV (Table EV1).

Informed consent was obtained from all study participants. The experiments conformed to the principles set out in the WMA Declaration of Helsinki and the Department of Health and Human Services Belmont Report. No sample size estimation was performed. Experiments were not blinded as samples were chosen based on their infection or vaccination status.

## Flow cytometry-based cell sorting

Cryopreserved PBMC samples were thawed and counted using a Vi-Cell XR Cell Viability Analyser (Beckman Coulter, CA, USA). Cells were then blocked with TruStain FcX™ PLUS for 10 min on ice

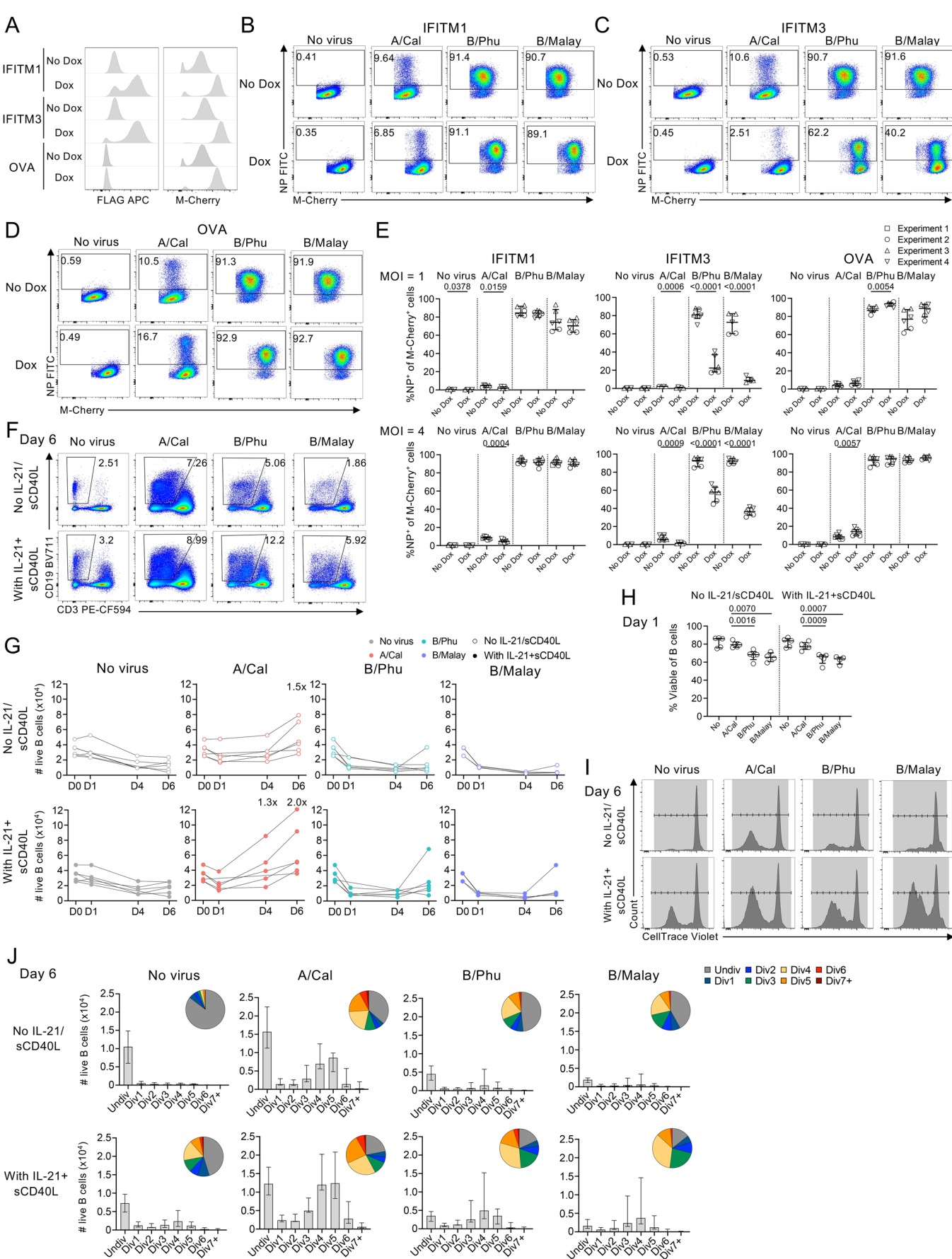

**Figure 8.  Function of the IFITM family during influenza virus infection.**

(A–E) A549 cells with DOX-inducible expression of FLAG-tagged IFITM1, IFITM3 and non-FLAG-tagged ovalbumin (OVA) were infected with influenza viruses for 8 h. (A) Representative FACS plots of DOX-inducible A549 cell lines. (B–D) Representative FACS plots and (E) frequency of NP$^+$ A549 cells with inducible levels of flag-tagged IFITM1, IFITM3 or non-flag-tagged control protein cytoplasmic OVA at 8 h post-infection from three independent experiments (4 for IFITM1 and OVA MOI = 4), each with two technical replicates. (F–J) Data following B cell proliferation assay from two independent experiments with biological replicates for no virus, A/Cal and B/Phu: $n_{D0} = 6$, $n_{D1} = 5$, $n_{D4} = 5$, $n_{D6} = 6$; and for B/Malay, $n = 4$ for all timepoints. PBMCs were stained with CellTrace Violet, followed by infections with influenza viruses at MOI = 4. Cells were cultured for 1, 4 and 6 days with or without IL-21 and soluble CD40 ligand (sCD40L). (F) Representative FACS plots of live B cells at day 6 post-infection. (G) Numbers of live B cells at 0, 1, 4 and 6 days post-infection with and without IL-21 and sCD40L. (H) Frequency of viable B cells at day 1 post-infection. (I) Representative histogram plots of B cell divisions at day 6 post-infection. (J) Numbers (bar graphs) and frequencies (pie charts) of live B cells at each division at day 6 post-infection. In (E, H), bars indicate the median with IQR. Statistical significance was determined with a two-tailed paired *t*-test. Exact *P* values are shown >0.0001 and <0.05. Source data are available online for this figure.

(Biolegend, CA, USA) before adding antibody cocktails containing fluorescent monoclonal antibodies, CITE-Seq antibodies and in-house A/California and B/Phuket recombinant HA (rHA) probes (Koutsakos et al, 2018) conjugated to streptavidin-PE barcoded oligos (Biolegend), as detailed in the Reagents and Tools table. This step was immediately followed by the addition of 1 μL of barcoded hashtag oligo (HTO, TotalSeq™-C025x anti-human Hashtag x Antibody, where x = 1 to 4, Biolegend) before incubation on ice for 30 min. Cells were washed, then bulk sorted on a SY3200 Cell Sorter (Sony, Tokyo, Japan) for CD19$^+$ rHA probe$^+$ B cells and CD3$^+$CD4$^+$CXCR5$^+$ Tfh cells (Appendix Fig. S2A). The A/California HA-probe was engineered with a Y98F mutation in the receptor-binding site to abolish binding to cell-surface sialic acids (Whittle et al, 2014). For the B/Phuket probe, the unmodified HA ectodomain was cloned, which already has an F at that position. In our previous studies, we have extensively validated and used our HA-probes from both A/California and B/Phuket strains (Hensen et al, 2021; Koutsakos et al, 2018; Nguyen et al, 2021; Zhang et al, 2023), including validations using two-colour labelling to demonstrate HA probe specificity (Tan et al, 2019).

## ScRNA-sequencing and quality control

For each Gel Beads-in-emulsion (GEM) reaction, sorted rHA probe$^+$ B cells and Tfh cells were combined, aiming for 22,000 total cells per reaction to input into the Chromium Controller using Chip A to achieve 10,000 cells per reaction at final throughput. GEM generations, cDNA amplification and gene expression (GEX)/TCR/BCR/CITE-Seq library generations were performed using the 10x Genomics Chromium Single Cell 5' Library & Gel Bead Kit (16 rxns PN-1000006, 10x Genomics, CA, USA) as per manufacturer's protocol CG000186 Rev D, before sequencing at Hartwell Center for Biotechnology, St. Jude Children's Research Hospital (Memphis, USA). All GEM reactions had high valid HTO barcode rates (>82.9%), sequencing saturation rates (>87.1%) and recovery rates of 32,502 total cells from four reactions (average 8126 cells/reaction; range 6152–9230 cells/reaction) (Appendix Fig. S2B). As routine quality control steps (Appendix Fig. S2C–E), we excluded cells with >0.1% mitochondrial genes, indicating dying cells, before regressing out cell cycle genes as described (Tirosh et al, 2016). The 28,028 cells that passed quality control were included in the following analyses.

## Analyses of scRNAseq data

Cells from unknown samples due to inconsistent HTO level were demultiplexed using the Vireo programme (Huang et al, 2019).

ScRNAseq data were analysed in R (v4.3.1) using the Seurat package (v4.3.0.1) (Hao et al, 2021). Blueprint database was used to annotate B cell phenotype subsets based on correlations of gene expression with the database using the SingleR package (v1.0) (Aran et al, 2019; Stunnenberg et al, 2016). For visualisation, UMAP, scatter plots, stacked bar plots, violin plots and bubble plots were also generated using the Seurat package along with ggplot2 (v3.4.3). The DEG comparisons between UMAP cell clusters and between different infection/vaccination timepoints were performed using the "FindMarkers" function of the Seurat package. Volcano plots were generated to visualise the DEGs using the EnhancedVolcano R package (v1.18.0). DEGs were calculated using the Wilcoxon rank-sum test with *P* values adjusted with a Bonferroni correction using all genes in the dataset. Metascape (v3.5.20230623) (Zhou et al, 2019) was used to perform function enrichment analyses, including gene ontology. For BCR data, raw fastq data were analysed using the Dandelion Python package (v0.3.3) for V(D)J analysis (Suo et al, 2023). Clonal expansions of B cells ( > 1 clonotype) were calculated using the Immcantation R package (v0.3.2) (Gupta et al, 2015). "HierarchicalClones" function of the Scoper sub-package was used for B cell clonal definition by hierarchical clustering with paired heavy and light chain sequences (Gupta et al, 2015). Circos plots and Venn diagrams were plotted using R packages circlize (v0.4.16) (Gu et al, 2014) and ggVennDiagram (v1.5.2) (Gao et al, 2021), respectively.

## Influenza viruses

Reverse-engineered IAV [A/H1N1/California/7/2009 (6 PR8 internal genes: 2 Cal/09 HA and NA genes)] and IBV (B/YAM/Phuket/3073/2013 and B/VIC/Malaysia/2506/2004) were sourced from Seqirus (CSL, Melbourne, Australia).

## Plaque assay

Standard plaque assays were performed using Madin-Darby canine kidney (MDCK) cells (ATCC, CCL-34) to determine viral titres in viral stocks for MOI calculations as previously described (Ng et al, 2010). The cell line tested mycoplasma negative, but were not authenticated. Briefly, samples were serially diluted and added to confluent MDCK cell monolayers in six-well plates for 45 min at 37 °C/5% $CO_2$. Cells were then overlaid with Leibovitz's L15 media (Gibco) supplemented with 0.01 M 4-(2-hydroxyethyl)-1-piperazineethanesulfonic acid (Hepes) buffer, 0.028% (w/v) $NaHCO_3$ (APS Finechem, NSW, Australia), 50 IU/mL penicillin (Thermo Fisher Scientific), 50 μg/mL streptomycin (Thermo Fisher Scientific) and 0.1% (w/v) of *n*-Tosyl-l-phenylalanine chloromethyl ketone (TPCK)-treated trypsin

(Worthington Biochemical Corporation, Lakewood, NJ, USA) mixed with 0.9% (w/v) agarose (Sigma Chemicals Co.). The plates were incubated at 37 °C/5% $CO_2$ for 3 days, and plaque counts were recorded as plaque-forming units (PFU).

## Influenza virus infection assay

The influenza virus infection assay was performed as previously described (Nguyen et al, 2021). Briefly, thawed PBMCs (<1.5e6) were cultured in serum-free media with influenza viruses at a multiplicity of infection (MOI) of 0, 1, 4 or 10 for 1 h at 37 °C/5% $CO_2$ using 48-well plates. Heat-inactivated FCS was then added (10% $V_f$) to wells and incubated for up to 8 or 22 h total incubation time. Cells were then harvested, washed, transferred into 96U-well plates, then stained with Live/Dead Aqua (#L34966, Thermo Fisher, Waltham, MA, USA), followed by surface and intracellular staining with panels outlined in the Reagents and Tools table. For intracellular staining, cells were permeabilised using the BD Cytofix/Cytoperm Kit (#554714, BD). Acquisition was on an LSRII Fortessa (BD). In some experiments at 1 h, heat-inactivated plasma was used instead of FCS from healthy individuals and patients with acute IAV, IBV, COVID-19 or RSV infection.

## A549 cell infection assay

Parental A549 cells were sourced from ATCC (CCL-185). The cell line tested mycoplasma negative, but were not authenticated. Parental A549 cells and A549 cell lines with DOX-inducible expression of FLAG-tagged IFITM1, IFITM3 and non-FLAG-tagged cytoplasmic ovalbumin as a control protein (Meischel et al, 2021) were infected with influenza viruses at MOI of 0, 1, 4 or 10 in serum-free media at 37 °C/5% $CO_2$. Doxycycline (DOX) (1 µg/mL, #D9891-5G, Sigma-Aldrich, MO, USA) was added 24 h prior to infection to induce overexpression of proteins. Expression of mCherry and intracellular FLAG (#637308, BioLegend) was assessed by FACS as a measure of transduction efficiency and expression of IFITM1 and IFITM3, respectively. Following 1 h infection, A549 cells were washed and cultured in serum-free media for a total of 8 hr. A549 cells were then harvested and stained for Live/Dead NIR (#L34976, Thermo Fisher) and IAV/IBV NP FITC (#GTX36902-500UL, GeneTex, CA, USA; #MA1-7306, Thermo Fisher). Acquisition was on an LSRII Fortessa.

## B-cell proliferation assay

B cell proliferation assay was performed as previously described (Auladell et al, 2019). Thawed PBMCs were stained with Invitrogen™ CellTrace™ Violet (CTV; #C34557, ThermoFisher). In brief, PBMCs were thawed and stained at 10e6 cells/mL with 1 µM CTV in warm PBS for 13 min at 37 °C before washing with RPMI/10% FCS. Stained cells (≤1e6/test) were subsequently infected with influenza viruses at MOI = 4 or incubated in serum-free medium for 1 h. Cells were cultured for 1, 4 or 6 days in the presence or absence of recombinant human IL-21 (50 ng/mL; #200-21-10UG, PeproTech, Rocky Hill, CT, USA) and soluble CD40 ligand (1 µg/mL; #310-02-50UG, PeproTech) then harvested and stained with Live/Dead NIR (#L34976, Thermo-Fisher), followed by surface and intracellular staining with panels outlined in Reagents and Tools table. For intracellular staining, cells were permeabilised using the BD Cytofix/Cytoperm Kit (#554714). Acquisition was on an LSRII Fortessa.

**The paper explained**

**Problem**

Despite the availability of vaccines, influenza virus infections remain a major cause of illness and death globally. Some vaccinated individuals still experience severe or even fatal influenza virus infections; however, the specific immune responses triggered by vaccination compared to those elicited by natural influenza virus infections are not fully understood. This knowledge gap limits our ability to develop more effective vaccines and treatments.

**Results**

To address this, we investigated the transcriptomic profiles of influenza-specific B-cells in vaccinated individuals and hospitalised patients infected with influenza A or B viruses (IAV, IBV). We found that interferon-stimulated gene signatures, particularly IF44L, IFITM1 and XAF1, were elevated in B-cells during acute IBV infections, but not in IAV-infected patients or vaccinees. We also observed that vaccination led to phenotypic changes and isotype class-switching in HA-specific B-cells, with evidence of clonal sharing between memory and atypical B cells. In vitro experiments further revealed that, compared to IAV, IBV infects immune cells more efficiently and may inhibit B-cell proliferation via the downstream effects of IFITM1.

**Impact**

Our findings provide important insights into the distinct transcriptomic profiles of influenza-specific B-cells in IAV and IBV infections and after vaccination. These results offer valuable information for improving vaccines and therapies to elicit protective B-cell responses and reduce adverse effects associated with natural infection.

## Graphics

Figure 1A and the synopsis were created with BioRender.com.

## Data availability

The datasets produced in this study are available in the following database: B cell sequencing data: Gene Expression Omnibus, accession number GSE309140, token krwvgesepdatnkp (https://www.ncbi.nlm.nih.gov/geo/query/acc.cgi?acc=GSE309140).

The source data of this paper are collected in the following database record: biostudies:S-SCDT-10_1038-S44321-026-00395-8.

## Peer review information

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

## Acknowledgements

We thank Lachlan Dryburgh for his help with data analysis. We thank the Melbourne Cytometry Platform for their services. We acknowledge Marco Herold for the lentiviral constructs for inducible overexpression. We thank all the patients and individuals who gave blood to the study. We thank all the Research Nurses at Alfred Hospital and Austin Health who helped with patient recruitment. This research was funded in whole or part by the National Health and Medical Research Council Investigator Grants: L2 to KK (#2033783), EL1 to THON (#1194036), L3 to SJK (#2016491) and L1 to AKW (#2026762); and Ideas Grants to BG (#2012463 and # 2038242). This work was supported by the Theme-based Research Scheme of the Research Grants Council of the Hong Kong Special Administrative Region (HKSAR), China (#T11-712/19-N) to KK, NIAID grant 1U01AI144616-01 to PGT and KK, and NIH contract CIVC-HRP (HHS-NIH-NIAID-BAA2018) to PGT and KK. For the purposes of open access, the author has applied a CC BY public copyright licence to any Author Accepted Manuscript version arising from this submission.

## Author contributions

**Wuji Zhang**: Conceptualisation; Data curation; Formal analysis; Validation; Investigation; Visualisation; Methodology; Writing—original draft; Writing—review and editing. **E Kaitlynn Allen**: Investigation; Methodology; Writing—review and editing. **Shihan Li**: Data curation; Formal analysis; Visualisation; Writing—review and editing. **Ilariya Tarasova**: Data curation; Formal analysis; Visualisation; Writing—review and editing. **Rubaiyea Farrukee**: Resources; Validation; Writing—review and editing. **Lukasz Kedzierski**: Investigation; Writing—review and editing. **Brad Gilbertson**: Validation; Investigation; Writing—review and editing. **Hayley A McQuilten**: Data curation; Formal analysis; Visualisation; Writing—review and editing. **Jennifer R Habel**: Methodology; Writing—review and editing. **Lilith F Allen**: Investigation; Writing—review and editing. **Steven Rockman**: Resources; Writing—review and editing. **Sarah L Londrigan**: Conceptualisation; Resources; Validation; Methodology; Writing—review and editing. **Stephen J Kent**: Resources; Writing—review and editing. **Adam K Wheatley**: Resources; Writing—review and editing. **Jason A Trubiano**: Resources; Writing—review and editing. **Tom C Kotsimbos**: Resources; Writing—review and editing. **Allen C Cheng**: Resources; Writing—review and editing. **Jan Schroeder**: Data curation; Formal analysis; Visualisation; Writing—review and editing. **Jeremy Chase Crawford**: Data curation; Formal analysis; Visualisation; Methodology; Writing—review and

editing. **Paul G Thomas**: Conceptualisation; Resources; Data curation; Formal analysis; Supervision; Funding acquisition; Visualisation; Methodology; Writing —review and editing. **Katherine Kedzierska**: Conceptualisation; Resources; Data curation; Supervision; Funding acquisition; Visualisation; Methodology; Writing—original draft; Project administration; Writing—review and editing. **Thi H O Nguyen**: Conceptualisation; Data curation; Supervision; Funding acquisition; Investigation; Visualisation; Methodology; Writing—original draft; Project administration; Writing—review and editing.

Source data underlying figure panels in this paper may have individual authorship assigned. Where available, figure panel/source data authorship is listed in the following database record: biostudies:S-SCDT-10_1038-S44321-026-00395-8.

## Disclosure and competing interests statement

HAMQ is a consultant for Ena Respiratory Pty. Ltd. KK received honoraria from Pfizer. PGT is on the SAB of Immunoscape and Cytoagents, consulted for JNJ, received travel support/honoraria from Illumina and 10x Genomics, and has patents related to TCR discovery. JCC and PGT have patents related to treating or reducing the severity of viral infections, including SARS-CoV-2. The remaining authors declare no competing interests.

