## [Peer Review File · EMBO Molecular Medicine]

IFN-gene signatures in B cells following influenza A and B virus infection and influenza vaccination

Thi Nguyen, Wuji Zhang, Emma Allen, Shihan Li, Ilariya Tarasova, Rubaiyea Farrukee, Lukasz Kedzierski, Brad Gilbertson, Hayley McQuilten, Jennifer Habel, Lilith Allen, Steven Rockman, Sarah Londrigan, Stephen John Kent, Adam Wheatley, Jason Trubiano, Tom Kotsimbos, Allen Cheng, Jan Schroeder, Jeremy Crawford, Paul Thomas, and Katherine Kedzierska

Corresponding authors: Thi Nguyen (thonguyen@unimelb.edu.au) , Katherine Kedzierska (kkedz@unimelb.edu.au)

Review Timeline:

Submission Date:	14th Dec 25
Editorial Decision:	19th Dec 25
Revision Received:	23rd Dec 25
Editorial Decision:	22nd Jan 26
Revision Received:	27th Jan 26
Accepted:	9th Feb 26

Editor: Zeljko Durdevic

Transaction Report:

Please note that the manuscript was previously reviewed at another journal. As EMBO Press has a transfer agreement with that journal, revision was invited based on the reports from that previous external submission.

RESPONSE TO REVIEWERS' COMMENTS

We sincerely thank both Reviewers for their insightful comments and interest in our study. We have improved the manuscript by providing further clarity on the importance of our findings after addressing the Reviewers' comments as outlined in a point-by-point form. We hope we have addressed the Reviewer's comments to their satisfaction.

We also thank the [other journal] Editor [redacted] for recommending the transfer of our manuscript to the *EMBO Molecular Medicine* and the *EMBO Molecular Medicine* Editor Zeljko the opportunity to consider our manuscript at the *EMBO Molecular Medicine* as a Resource article.

Reviewer #1 (Remarks to the Author):

The investigators analyze patterns of interferon gene stimulation in influenza hemagglutinin (HA) specific B cells in hospitalized patients during acute infection with influenza A virus (IAV) or influenza B virus (IBV) as well as at 1 month convalescence. An interferon gene signature was uncovered for acute IAV infection which included activation of IF44L, IFITM1 and XAF1 in total B cells. Notably this pattern was not seen when patients were infected with IAV or IAV + IBV. Differences between IBV and IAV infectivity on PBMC were also observed in vitro and plasma from patients that were hospitalized for COVID-19, IBV, IAV also increased infectivity of both viruses. IBV infection of PBMC reduced B cell proliferative capacity to a greater extent than did IAV, possibly correlating with the antiproliferative effect of IFITM1.

We thank the Reviewer for an accurate summary of our manuscript.

General:

Technically the work is well executed and is of high caliber.

We thank the Reviewer for the appreciation of the high caliber of our study and recognizing the technical soundness of our work.

However, the large-scale gene expression and clonal mapping across B cell subsets within the human infection studies, while extensive and impressive, are open ended and seem to be hypothesis generating rather than hypothesis testing. Multiple parameters change post immune challenge with notable expectations being fulfilled (e.g. B cell expression of CXCR4 or activation of effector status in response to IIV), but this reviewer is left wondering what the truly novel mechanistic insight is being gained.

We agree with the Reviewer and acknowledge that we have provided a more hypothesis driven data set, which Reviewer #2 also agreed as being a "*wealth of information...that could be highly interesting to the field.*" As such, following the *Nature Microbiology* Editor's suggestion, we have transferred our manuscript to *EMBO Molecular Medicine* for consideration as a Resource article. We believe our comprehensive dataset on human B cell subsets during hospitalized viral infection and following vaccination will be a valuable resource for future mechanistic studies, particularly surrounding the role of IFITM family proteins on B cell responses.

It seems that fundamental questions could be asked with respect to B cell repertoire sharing and clonality tracking between subsets etc, but a clear biological question is also not being formulated in this area.

We have now added fundamental questions for further studies stemming from our dataset as follows:

Abstract (page 2):

“We provide key insights into B-cell immunity towards IBV and IAV infections and vaccination, which will inform rational vaccine design and therapeutic strategies aimed at eliciting robust HA-specific B-cell responses, while minimizing adverse effects caused by natural infection.”

Introduction (page 4):

“These findings have important implications to inform future vaccine and treatment strategies against influenza virus infections. Specifically, targeting the transcriptomic profiles of HA-specific B cells may help to elicit more robust B cell responses.”

Discussion (pages 15-16):

“Altogether, our findings provide much needed knowledge for detailed understanding of immune responses against IAV and IBV infections and vaccination responses. Our dataset provides proof-of-concept for exploring the link between HA-specific B cell clonality and transcriptomic signatures, as well as underscoring the need for further studies on the role of IFITM family members in B cell responses.”

The difference in interferon stimulation between IAV and IBV in B cells in vivo is interesting, but it is hard to envision how the ‘corresponding’ PBMC infection studies, which don’t reflect flu infection of respiratory mucosa, provide a compelling correlate.

While we have conducted *in vitro* investigations to examine the role of IFITM family members in B cells following influenza virus infection, we acknowledge that further studies are required to fully elucidate the mechanistic insights regulating IFN-stimulated genes in B cells, particularly within the respiratory mucosa (such as using lung organoid cells) and to explore the relationships between B cell subsets with distinct gene expression, B cell repertoire, and its clonality. These future studies are beyond the scope of the current manuscript.

We have added this limitation to our Discussion as follows (page 15):

“We acknowledge that our PBMC infection studies do not entirely reflect influenza virus infection of respiratory mucosa, therefore, future in vitro studies harnessing lung organoid cells is warranted.”

Other

Figure 1C. Typically two-color HA labeling is performed to ensure B cell binding independent of color.

We thank the Reviewer for this question. Previously, we have extensively validated and used our HA probes from both A/California and B/Phuket strains in our published works (Zhang W *et al.* Clinical & Translational Immunology 2023; Nguyen THO *et al.* Nature Communications *et al.* 2021; Koutsakos M *et al.* Science Translational Medicine 2018; Hensen L *et al.* PNAS 2021), including validations using two-colour labelling to demonstrate HA probe specificity (Tan H-X *et al.* J Clin Invest 2018). As we were limited by the fluorochromes used in our study, we opted to use single-color HA probes, as after cell sorting, we further confirmed HA specificity by combining manual gating based on probe specificity and B cell receptor clonality analyses. We agree that dual-staining should be performed for newly made HA probes for validation, as well as HA probes that tend to bind

non-specifically.

Following the Reviewer's comment, we described the previous validations of our HA probes in Methods (page 18):

“In our previous studies, we have extensively validated and used our HA probes from both A/California and B/Phuket strains (Zhang W et al. Clinical & Translational Immunology 2023; Nguyen THO et al. Nature Communications et al. 2021; Koutsakos M et al. Science Translational Medicine 2018; Hensen L et al. PNAS 2021), including validations using two-colour labelling to demonstrate HA probe specificity (Tan H-X et al. J Clin Invest 2018).”

The authors should also state whether their probes were attenuated for sialic acid binding.

The A/California probe was attenuated for sialic acid binding by a Y98F mutation in the receptor-binding site (Whittle et al J Virol 2014), but not the B/Phuket probe which already has a F at that position. We have now clarified in the Methods section as follows (page 18):

“The A/California HA probe was engineered with a Y98F mutation in the receptor-binding site to abolish binding to cell-surface sialic acids (Whittle et al J Virol 2014). For the B/Phuket probe, the unmodified HA ectodomain was cloned which already has a F at that position.”

There is a general trend to equate VH gene usage with specific broadly neutralizing activity; e.g. IGHV1-69 with influenza stem bnAbs or IGHV3-30 as hardcoding for flu B. These are broad oversimplification and should not be done. E.g. IGHV1-69 bnAb precursors require the VH alleles displaying F54 in CDRH2 and also enrichment of Tyr at Kabat positions 98-100 in the CDRH3.

We thank the Reviewer for their insightful comments and have now removed such statements within the text and just focussed on our main BCR gene usage findings within the dataset and similar observations with our previous study. All comparisons to the literature have been removed from the text.

The revised paragraphs relating to the “BCR gene segment...” Results section now reads as follows (page 9):

*“While overall BCR segment usage was diverse, heavy chains IGHV3-23 (light yellow) and IGHV3-30 (yellow) were frequently detected in A/California HA-specific B cells from both vaccinees at all timepoints (8-22%, except that IGHV3-30 (yellow) was not detected for VAX1 at d0), with two clonotypes detected across multiple timepoints in VAX2 (**Fig. 4b-d, Extended Data Fig. 6a-c, Supplementary Table 2**). For VAX1, various IGHV gene segments were mainly paired with IGLV2-8 (cyan) or IGKV3-15 (light teal), including expanded clonotypes (IGHV1-46/IGLV2-8, IGHV3-7/IGKV3-15 and IGHV3-23/IGKV3-15). Similarly, two light chain segments, IGLV2-8 (cyan) and IGKV3-15 (light teal), were detected in VAX2 with diverse heavy chains but were less clonally expanded than VAX1.*

*For B/Phuket HA-specific B cells, heavy chains IGHV3-30 (yellow) and IGHV3-30-3 (golden yellow) were observed in both donors at all time points with the most seen at d28 with multiple IGLV pairings, representing 7.2% (57 out of 796) of our dataset (**Supplementary Table 2**). Similarly, we observed ~25% of B/Phuket HA-specific B cells having IGHV3-30 or IGHV3-30-3 in our previous IIV study (Koutsakos et al STM 2018). On d28 post vaccination, other variable gene pairings such as IGHV5-10-1/IGKV3-20 and IGHV3-15/IGLV6-57 in VAX1, and IGHV3-7/IGKV3D-20 and IGHV3-15/IGKV3-11 in VAX2 were also commonly observed (**Fig. 4b-d, Extended Data 6a-c**).”*

Reviewer #2 (Remarks to the Author):

In their manuscript, Zhang et al. characterize the B cell responses in people either vaccinated against influenza or infected with influenza. They include influenza A virus (IAV) and influenza B virus (IBV). Their study provides a wealth of information on the B cell responses that could be highly interesting to the field. However, there are a number of limitations that limit my enthusiasm. First, the authors included only hospitalized IAV and IBV patients, many of whom seem to be on immunosuppressive treatment. Thus, they are only looking at a subset of infections and should not generalize their findings to infection as compared to vaccination. Second, the authors aim to follow up on their results from analysis of patient B cells in an *in vitro* system where they infect PBMCs but it is unclear if patient B cells are infected *in vivo*. If they are not infected I do not see how figures 6-7 link to the rest of the study. Lastly, they add data on IFITM sensitivity of IBV versus IAV in A549 cells but again it is unclear how this relates to the patient data in B cells.

We thank the Reviewer for their insightful comments and as such, we have greatly improved the manuscript by removing any comparative statements between infection and vaccination and for providing better clarity and rationale for the *in vitro* experiments in Figures 6-8.

To summarise for the Reviewer, our *in vitro* findings enabled direct comparisons of different infection conditions by using the same set of PBMCs. We showed that there were differences in NP+ staining of B cells and other immune cell subsets between IAV and IBV infection, which modestly increased for both viruses using patient plasma from patients hospitalized for COVID-19, IBV, IAV and RSV in the culture medium compared to using healthy plasma. Given the high gene expression levels of IFITM family members *IF44L*, *IFITM1* and *XAF1* in B cells from our scRNA-seq data for IBV-infected patients, we then aimed to use an epithelial A549 cells with dox-inducible IFITM proteins, to further dissect out the potential roles of these IFITM family members in the context of IAV and IBV infection, noting that they are different to B cells. We also found that IBV infection of healthy PBMCs reduced the B cell proliferative capacity to a greater extent than IAV, which was possibly associated with the antiproliferative effect of IFITM1, where *IFITM1* gene expression was highly observed in our IBV-patient B cells. Although we cannot directly compare the *ex vivo* scRNA-seq data to the *in vitro* work, we believe this is still important to include as a Resource article, as suggested by our discussions with the Editors’.

Major points:

- In the title and abstract the authors claim that they compare influenza infection-induced with influenza vaccination-induced B cell responses, but they are only looking at a very specific subset of infections, namely in hospitalized patients, many of whom seem to be on immunosuppressive treatment and only H1N1 infections are included. I appreciate that getting human samples is not easy, but this subgroup is not representative of “influenza infection” as claimed by the authors. Most infections result in mild disease, and this is not included at all. I do not see how the samples from the vaccinated can be directly compared to the samples from the infected in this study. This is also evident from figures 2-4 being only on the vaccinated samples, whereas the remaining figures focus only on infected samples.

We thank the Reviewer for their insightful comments and acknowledge that our study focuses on a limited cohort comprising influenza vaccinees and patients hospitalized with IAV or IBV infection. We have now removed statements of comparisons between infection and vaccination throughout the manuscript, particularly in the Title and Abstract as follows:

Title (page 1):

“IFN-gene signatures in B cells following influenza A and B virus infection and influenza vaccination”

Abstract (page 2):

“Yet our understanding of immune features elicited by vaccination and influenza A and B virus infection (IAVs, IBVs) is limited.”

And

“We provide key insights into B-cell immunity towards IBV and IAV infections and vaccination, which will inform rational vaccine design and therapeutic strategies...”

- L. 120, 123: Groups IAV1, IAV2, ... VAX1 etc. are not defined or described well when first mentioned. This information is crucial but currently unclear. The authors should clearly describe how the demographics (incl. underlying diseases, medication etc.) compare between the infected and vaccinated groups.

We have now provided further participant demographics from Supplementary Table 1 into the main text for better clarification of the two cohorts as follows (page 4):

“Acute samples from hospitalized influenza-infected patients (IAV1, IAV2, IBV1 and IBV2 patients, aged 21-63 yrs, all female) were collected at 5-18 days post disease onset (between 2-4 days post hospital admission), while follow-up samples were collected following hospital discharge at 27-51 days post disease onset (**Fig. 1a**). Healthy influenza vaccinees (VAX1 and VAX2 participants, aged 25-27 yrs, both female) were blood sampled on day 0 pre-vaccination and on d7 and d28 post-vaccination (**Fig. 1a**). Patients were all admitted to the respiratory/general ward, with hospital stays ranging between 2-7 days, and all survived to 30 days post-admission. IAV2 required non-invasive oxygen support, while all except IAV1 received oseltamivir treatment. IAV1 and IBV2 were considered highly susceptible to severe influenza disease due to underlying chronic respiratory disease and immunosuppression. IBV2 also had cardiac disease and chronic renal disease (**Supplementary Table 1**).”

- Fig. 6: It is unclear to me how the *in vitro* infections of PBMCs compare to the *in vivo* situation. Do the authors have evidence that B cells are infected *in vivo* or is this an artefact from the *in vitro* system? The scRNA seq data should reveal viral transcripts if B cells are infected *in vivo*. If they are not infected *in vivo* this part of the manuscript does not link to the rest.

We thank the Reviewer and acknowledge that the *in vitro* infections of healthy PBMCs does not fully replicate the conditions within the patients' blood samples, but an extension of our studies following on from the scRNAseq findings. We do not have clear evidence that the patient B cells were infected *in vivo* as we did not detect any viral transcript from our scRNA-seq data, however this may be due to the later sampling timepoints which was between 5-18 days post disease onset.

We have better clarified the rationale for our *in vitro* PBMC infection experiments and toned down statements linking our *ex vivo* data to our *in vitro* data throughout the text as follows:

Results page 10:

“Given we observed enhanced activation of the IFN signalling pathway in B cells following IBV infections, we next asked whether there were any virus-mediated differences between IAV and IBV infection by utilizing an *in vitro* influenza virus infection assay, which incubated healthy PBMCs with

live IAV and IBV viruses at MOI of 1, 4 and 10 to assess *infectivity rates* in lymphocyte subsets and monocytes *in a controlled in vitro setting*.”

Results page 12:

“Overall, our human *in vitro* assays provide *further* insights into the transcriptomic differences observed in the B cell response following IAV and IBV infections. IBV infections, at least in our model system, resulted in higher infectivity rates in B *cells and* lower B cell viability and proliferative capacity. Our findings may *partially* explain why we observed such strong upregulation of IFN signalling genes required for recovery from acute B/Phuket-infected patients, compared to our A/California-infected patients.”

Discussion page 14:

“Therefore, although our dataset shows a strong IFN signature following IBV infection, *we have no clear evidence whether these B cells were “more” infected in vivo as we did not detect any viral transcript from our scRNA-seq data. However, we also cannot rule this hypothesis out as any viral transcript may have been undetectable due to the later sampling timepoints which was between 5-18 days post disease onset.*”

- Fig. 6d lacks an essential control: The authors need to measure viral titres at 1-2h post infection to see how much of the virus measured at 8h is left-over from the input. Only if this control is available and titres at 8h are higher than at 1-2h one can conclude on virus growth.

We thank the Reviewer for their comment, and they are correct to point out that conclusions relating to virus growth cannot be made without baseline viral titres at 1-2 hours post-infection to account for viral inoculum. The main purpose of the data in Figure 6 was to measure viral infectivity by intracellular NP staining of the PBMC subsets at 8h to ensure that the cell types in question were susceptible to virus infection and supported at least the early stages of viral replication including the synthesis of new viral NP. The data show an increasing percentage of influenza-infected cells (based on NP expression) that correlates with increasing MOI used for infection (Figure 6a-c). As such, the plaque assay performed at 8 and 22 hours relating to virus release (Fig. 6d: virus growth) is unnecessary and has been removed to aid clarity. As such, we have also removed Figure 6e correlating NP staining with viral titres.

- L. 369-370 “These findings suggest that the elevated NP+ cell frequencies that we observe may be driven by the plasma mediators rather than by virus-specific factors.” I am not following how the authors link the results from figure 6 with the results in figure 7. In fig. 6 they see increased infection levels in B cells (and a few other immune cells) for IBV compared to IAV when using PBMCs from healthy donors. If the effects (higher NP+ frequencies for IBV compared to IAV) are plasma-mediated why do they see higher infection levels in fig. 6? I would argue that it is virus-mediated given that one can see the effect without the addition of plasma (fig. 6) or with different donor plasmas (fig. 7).

We thank the Reviewer for their insightful comments and we absolutely agree with the Reviewer. The differences in PBMC infectivity between IAV and IBV strains shown in Figure 6 are likely due to intrinsic properties of the viruses. In Figure 7, the increased infectivity with IBV still remained. The presence of patients’ plasma, with different respiratory viral infections including IAV, IBV, COVID-19, and RSV, only modestly increased the infectivity rates compared to healthy plasma.

We have better clarified the text relating to Figure 7 as follows:

Results page 11:

“To determine whether the inflammatory environment *added to the effects of the infectivity between IAV and IBV*, we additionally performed an *in vitro* influenza virus infection assay for 22 hours at *MOI=4*, but now using 10% heat-inactivated plasma obtained from patients hospitalized with either IAV, IBV, COVID-19 and RSV (n=20; **Fig. 7**). *Like using FCS in the cultures previously, IBV infection was still higher than IAV infection. However,* compared to healthy plasma (obtained from 8 healthy participants) culturing conditions, plasma from IAV and IBV patients *modestly* increased the frequency of NP⁺ B cells following IBV infections by 1.7-2.8-fold (**Fig. 7a**).”

And

“These findings suggest that, *in addition to virus-specific factors mainly contributing to the IBV>IAV hierarchy, plasma mediators within acute hospitalised patient samples can also contribute to modest increases in NP⁺ cell frequencies. Notably, that these plasma mediators are possibly shared across different respiratory viral infections including IAV, IBV, COVID-19 and RSV.*”

Discussion page 14:

“Interestingly, we observed a significantly higher frequency of NP⁺ B cells, T cells and NK cells following IBV infection compared to IAV infection. Plasma from patients with influenza, COVID-19 or RSV, *as compared to healthy plasma, modestly* increased the infectivity of IAV and IBVs *further*, likely attributable to the highly inflammatory milieu present in the plasma.

- Fig. 8: The authors study IAV and IBV inhibition by IFITMs in A549. It is unclear to me how the IFITM effects in A549 are relevant for observations in B cells. For mechanistic follow-up experiments in B cells would be required but these only make sense if clear evidence for infection of B cells in vivo is obtained. Moreover, even if IBV was less sensitive to inhibition by IFITM1 in B cells I do not see how this would explain why IBV infection in patients seems to lead to higher ISG levels in B cells compared to IAV infection.

Again, we thank the Reviewer immensely for their insightful comments and have now provided better rationale for using the IFITM inducible A549 cell lines, and highlighted the need to further investigate the role of IFITM family proteins in B cells as follows:

Results page 11:

“To investigate whether these ISGs have direct antiviral effects during IBV infection *in comparison to IAV infection*, we evaluated the impact of inducible IFITM1 and 3 protein overexpression on *IAV and IBV* infections using A549 cell lines in the presence of doxycycline.”

Results page 12:

“Although IFITM1 was more highly expressed compared to IFITM3 from our single-cell 10x data *on B cells*, it did not have a clear effect on reducing IBV infections *in the epithelial A549 cells, warranting further investigations of impact of IFITM proteins on B cells.*”

Discussion page 15:

However, although B cells in our scRNAseq data exhibited a rapid upregulation of ISGs, our novel findings from *the epithelial A549 cell lines* suggests that IFITM1 expression may not confer the same level of antiviral protection against IBV as it does against IAV, *warranting further investigation of these IFITM proteins in B cells.*”

And

“While IFITM1 did not demonstrate direct inhibition of IBV infection *at least in the A549 cells*, it may influence B cell proliferation.”

Further, as mentioned above for Figure 6, we have also toned down our statements linking heightened ISGs (*ex vivo* data) to higher IBV infectivity compared to IAV (our *in vitro* data) as follows (page 10):

“*Given we observed enhanced activation of the IFN signalling pathway in B cells following IBV infections, we next asked whether there were any virus-mediated differences between IAV and IBV infection by utilizing an in vitro influenza virus infection assay, which incubated healthy PBMCs with live IAV and IBV viruses at MOI of 1, 4 and 10 to assess infectivity rates in lymphocyte subsets and monocytes in a controlled in vitro setting.*”

Minor points:

- In the abstract and introduction, the authors state that their studies will aid future vaccine design. Can they be more specific? How would these results now change or inform vaccine design strategies? In my opinion, giving some details and specifics here would be important for the non-expert reader to understand the impact of the results.

We thank the Reviewer for their comment. We have now added further details on how our study may aid future vaccine design as follows:

Abstract (page 2):

“*We provide key insights into B-cell immunity towards IBV and IAV infections and vaccination, which will inform rational vaccine design and therapeutic strategies aimed at eliciting robust HA-specific B-cell responses, while minimizing adverse effects caused by natural infection.*”

Introduction (page 4):

“*These findings have important implications to inform future vaccine and treatment strategies against influenza virus infections. Specifically, targeting the transcriptomic profiles of HA-specific B cells may help to elicit more robust B cell responses.*”

- L. 100: What does “high-level” mean here?

We were referring to a more complex 10X experiment encompassing hash-tagging of samples, barcode labelled HA-probes and CITEseq antibodies, which we have now clarified as follows (page 3):

“*...we performed single cell RNA sequencing (scRNA-seq) including hash-tagging of samples, barcode-labelled HA-probes and CITEseq antibodies...*”

- In my opinion, fig. 2g and 3j do not add much and could be cut.

We have followed the Reviewer’s suggestions and removed Figures 2g and 3j.

- The graphs in fig. 4b-e are suitable to display the diversity in heavy and light chain usage, but it is difficult to see the overrepresented IGHV and IGLV/IGKV. Could the authors highlight some of them or at least refer to the respective colour in the text of the results section?

We have now labelled the overrepresented IGHV and IGLV/IGKV segments in the main Figures 4b and 4c (*please see below*) as well as included the colours in brackets in the Results section. Following Reviewer #1's comment on page 3 of Rebuttal, we have now condensed the Results section which reads as follows (page 9):

“While overall BCR segment usage was diverse, heavy chains IGHV3-23 (*light yellow*) and IGHV3-30 (*yellow*) were frequently detected in A/California HA-specific B cells from both vaccinees at all timepoints (8-22%, except that IGHV3-30 (*yellow*) was not detected for VAX1 at d0), with two clonotypes detected across multiple timepoints in VAX2 (**Fig. 4b-d, Extended Data Fig. 6a-c, Supplementary Table 2**). For VAX1, various IGHV gene segments were mainly paired with IGLV2-8 (*cyan*) or IGKV3-15 (*light teal*), including expanded clonotypes (IGHV1-46/IGLV2-8, IGHV3-7/IGKV3-15 and IGHV3-23/IGKV3-15). Similarly, two light chain segments, IGLV2-8 (*cyan*) and IGKV3-15 (*light teal*), were detected in VAX2 with diverse heavy chains but were less clonally expanded than VAX1.

For B/Phuket HA-specific B cells, heavy chains IGHV3-30 (*yellow*) and IGHV3-30-3 (*golden yellow*) were observed in both donors at all time points with the most seen at d28 with multiple IGLV pairings, representing 7.2% (57 out of 796) of our dataset (**Supplementary Table 2**).”

Modified **Figure 4b and c**, with key genes labelled on the circos plot:

- Fig. 5e: In my opinion, this graph does not add anything. The ISGs do not interact physically; they come up as network because their expression is regulated by IFN, a concept that is very clearly established and does not need to be shown with this graph.

We thank the Reviewer for this comment. We have now removed Fig. 5e.

- Fig. 6 is very crowded and thus difficult to follow. Did the authors consider moving panels in a and b to supplementary information to focus on panels c?

We did consider rearranging the figures but following the Reviewer's Major comment above related to virus growth, we have since removed Figures 6d-e which has made the figure look less dense.

- The figure legends have very little detail so that it is difficult to understand what was done in each experiment.

We have revised the figure legends for Figures 6-8 to provide more information for each experiment as follows (pages 27-28):

“**Figure 6. Influenza virus infection assay of PBMCs with IAV and IBV.** PBMCs were infected with A/California, B/Phuket, and B/Malaysia strains *in vitro* at MOI of 1, 4 and 10 for 8 hours or 22 hours ($n = 6$ per group). Intracellular NP staining was performed to investigate the proportion of cells that are NP positive. (a) Representative FACS plots of NP-staining on immune cell populations. (b-c) Frequencies of NP⁺ cells at (b) 8 hours and (c) 22 hours post-infection. Bars indicate the median with IQR. Statistical significance was determined with RM one-way ANOVA with Geisser-Greenhouse correction followed by Sidak multiple comparisons.”

“**Figure 7. Influenza virus infection assay co-cultured with plasma from patients hospitalized with IAV, IBV, COVID-19 and RSV.** The assay was performed as described in Figure 6 legend and Methods section at MOI=4 for 22 hours, with the exception that heat-inactivated patient plasma samples were added 1-hour post-infection of influenza viruses. Experiments from 4 buffy pack donors cultured with IAV ($n = 5$), IBV ($n = 5$), COVID-19 ($n = 5$), RSV ($n = 5$) and healthy plasma ($n = 8$), per virus condition. (a-g) The frequency of NP⁺ cells out of (a) B cells, (b) CD3⁺CD19⁺CD56⁻ cells, (c) CD4⁺ T cells, (d) CD8⁺ T cells, (e) MAIT cells, (f) $\gamma\delta$ -T cells, and (g) NK cells. Bars indicate the median with IQR. Statistical significance was determined with Ordinary one-way ANOVA followed by Turkey’s multiple comparisons.”

“**Figure 8. Function of IFITM family during influenza virus infection.** (a-e) Parental A549 cells and A549 cells with DOX-inducible expression of FLAG-tagged IFITM1, IFITM3, and non-FLAG-tagged ovalbumin (OVA) were infected with influenza viruses for 8 hours. (a) Representative FACS plots of DOX inducible A549 cell lines. (b-d) Representative FACS plots and (e) frequency of NP⁺ A549 cells with inducible levels of flag-tagged IFITM1, IFITM3 or non-flag tagged control protein cytoplasmic OVA at 8 hours post-infection from 3-4 independent experiments. (f-j) Data following B cell proliferation assay from 2 independent experiments ($n = 4-6$). PBMCs were stained with CellTrace Violet, followed by infections with influenza viruses at MOI=4. Cells were cultured for 1, 4 and 6 days with or without IL-21 and soluble CD40 ligand (sCD40L). (f) Representative FACS plots of live B cells at day 6 post-infection. (g) Numbers of live B cells at 0, 1, 4 and 6 days post-infection with and without IL21 and sCD40L. (h) Frequency of viable B cells at day 1 post-infection. (i) Representative histogram plots of B cell divisions at day 6 post-infection. (j) Numbers (bar graphs) and frequencies (pie charts) of live B cells at each division at day 6 post-infection. In e and h, bars indicate the median with IQR. Statistical significance was determined with a two-tailed paired t test.”

19th Dec 2025

Dear Dr. Nguyen,

Thank you for the submission of your revised manuscript to EMBO Molecular Medicine. We have now heard back from one referee who agreed to re-evaluate your manuscript. This referee also assessed author responses to concerns raised by the other referee. I am pleased to inform you that we will be able to accept your manuscript pending reformatting in accordance with the journal's requirements. Please check the points below as well as our Author Guidelines for more information <https://www.embopress.org/page/journal/17574684/authorguide#manuscriptpreparation>

I look forward to receiving a revised version of your manuscript as soon as possible.

Yours sincerely,

Zeljko Durdevic

Zeljko Durdevic
Senior Editor
EMBO Molecular Medicine

*** Instructions to submit your revised manuscript ***

When preparing your revised manuscript, please refer to our guidelines: <https://link.springer.com/journal/44321/submission-guidelines#cms-Revised-submissions>. We perform an initial quality control of all revised manuscripts before re-review; failure to include requested items will delay the evaluation of your revision.

We require:

- 1) A .docx formatted version of the manuscript text (including legends for main figures, EV figures and tables). Please make sure that the changes are highlighted to be clearly visible.
- 2) Individual production quality figure files as .eps, .tif, .jpg (one file per figure). For guidance, download the 'Figure Guide PDF': <https://media.springernature.com/original/springer-cms/rest/v1/content/27825798/data/v1>.
- 3) A .docx formatted letter INCLUDING the reviewers' reports and your detailed point-by-point responses to their comments. As part of the EMBO Press transparent editorial process, the point-by-point response is part of the Review Process File (RPF), which will be published alongside your paper.
- 4) A complete author checklist, which you can download from our author guidelines. Please insert information in the checklist that is also reflected in the manuscript. The completed author checklist will also be part of the RPF.
- 5) Please note that all corresponding authors are required to supply an ORCID ID for their name upon submission of a revised

manuscript.

6) It is mandatory to include a 'Data Availability' section after the Materials and Methods. Before submitting your revision, primary datasets produced in this study need to be deposited in an appropriate public database, and the accession numbers and database listed under 'Data Availability'. Please remember to provide a reviewer password if the datasets are not yet public.

7) For data quantification: please specify the name of the statistical test used to generate error bars and P values, the number (n) of independent experiments (specify technical or biological replicates) underlying each data point and the test used to calculate p-values in each figure legend. The figure legends should contain a basic description of n, P and the test applied. Graphs must include a description of the bars and the error bars (s.d., s.e.m.).

9) Our journal encourages inclusion of *data citations in the reference list* to directly cite datasets that were re-used and obtained from public databases. Data citations in the article text are distinct from normal bibliographical citations and should directly link to the database records from which the data can be accessed. In the main text, data citations are formatted as follows: "Data ref: Smith et al, 2001" or "Data ref: NCBI Sequence Read Archive PRJNA342805, 2017". In the Reference list, data citations must be labeled with "[DATASET]". A data reference must provide the database name, accession number/identifiers and a resolvable link to the landing page from which the data can be accessed at the end of the reference.

12) Author contributions: You will be asked to provide CRediT (Contributor Role Taxonomy) terms in the submission system. These replace a narrative author contribution section in the manuscript.

13) A Disclosure and competing interests statement should be provided in the main text.

14) Every published paper includes a 'Synopsis' to further enhance discoverability. Synopses are displayed on the journal webpage and are freely accessible to all readers. They include a short stand first (maximum of 300 characters, including space) as well as 2-5 one-sentences bullet points that summarizes the paper. Please write the bullet points to summarize the key NEW findings. They should be designed to be complementary to the abstract - i.e. not repeat the same text. We encourage inclusion of key acronyms and quantitative information (maximum of 30 words / bullet point). Please use the passive voice. Please attach these in a separate file or send them by email, we will incorporate them accordingly.

15) Include a Reagents and Tools Table as part of the Methods section, which can be downloaded from our author guidelines.

Photos 400-800 DPI

*Additional important information regarding figures and illustrations can be found at

<https://media.springernature.com/original/springer-cms/rest/v1/content/27825798/data/v1>

***** Reviewer's comments *****

Referee #1 (Comments on Novelty/Model System for Author):

The authors have appropriately addressed concerns of the reviewers.

The authors addressed the editorial issues.

22nd Jan 2026

Dear Dr. Nguyen,

Thank you for the submission of your revised manuscript to EMBO Molecular Medicine and please accept my apologies for the delay in getting back to you due to the holiday season and large number of submissions we experienced recently. I am pleased to inform you that we will be able to accept your manuscript pending the following final amendments:

1) In the main manuscript file, please do the following:

- Please address all comments suggested by our data editors listed below:

o Data availability statement:

1. Please note that the specific URL for GSE309140 dataset is not provided in the data availability statement.

o Figure legends:

1. Please note that the exact p values are not provided in the legends of figures 6C, 7A, B, C, E.

2. Please note that information related to n is missing in the legends of figures 1G, 2A-C, E, F; 5A, D.

- Remove "Summary" and leave only "Abstract".

- Rename "Materials and Methods" to "Methods".

- In Methods, provide the antibody dilutions that were used for each antibody.

- Remove BioRender reference from legends and add a dedicated section to the Methods.

Graphics:

(some of the... OR Figure #... OR synopsis) Graphics were created with BioRender.com.

- Indicate in legends number and nature of replicates and exact p= values, not a range, along with the statistical test used. To keep the figures "clear" some authors found providing an Appendix table Sx with all exact p-values preferable. You are welcome to do this if you want to.

- Please remove Reagents and Tools Table from the manuscript file and uploaded it as a separate .doc file.

- In data availability statement remove the sentence "All other data needed to support the conclusions of the paper are present in the paper or Appendix". Please use the following format to report the accession number of your data:

[data type]: [full name of the resource] [accession number/identifier] ([doi or URL or identifiers.org/DATABASE:ACCESSION])

- Correct the reference citation in the text and reference list. In the text, a reference should be cited by author and year of publication. Include a space between a word and the opening parenthesis of the reference that follows. In the reference list, citations should be listed in alphabetical order. Where there are more than 10 authors on a paper, 10 will be listed, followed by "et al.". Please check "Author Guidelines" for more information.

<https://www.embopress.org/page/journal/17574684/authorguide#referencesformat>

2) Tables: Please rename Table EV2 to Dataset EV1, add a legend to the excel file and update its callout in the main text.

3) Appendix: Please add page numbers to the table of contents.

4) Funding: Please merge it with Acknowledgments and remove the word "funding". Please add the detailed and comprehensive funding information to our system, including recipients and project numbers; this funding information will be linked to PubMed from the published article so it is important that the list in our system is complete and accurate. Please do not use the "Comments" field, which cannot be linked.

5) The Paper Explained: Please provide "The Paper Explained" and add it to the main manuscript text. Please check "Author Guidelines" for more information. <https://www.embopress.org/page/journal/17574684/authorguide#researcharticleguide>

6) Synopsis:

- Synopsis image: Please upload the image as a high-resolution .jpeg/.png file 550 px-wide x 300-600 pixels high.

- Synopsis text: Please upload it as a separate.doc file.

7) As part of the EMBO Publications transparent editorial process (see our Editorial at

<http://embomolmed.embopress.org/content/2/9/329>), EMBO Molecular Medicine will publish online a Review Process File (RPF) to accompany accepted manuscripts. This file will be published in conjunction with your paper and will include the anonymous referee reports, your point-by-point response and all pertinent correspondence relating to the manuscript. Let us know if you want to remove or not any figures from it prior to publication. Please note that the Authors checklist will be published at the end of the RPF.

8) Please provide a point-by-point letter INCLUDING my comments as well as the reviewer's reports and your detailed responses (as Word file).

I look forward to reading a new revised version of your manuscript as soon as possible.

Yours sincerely,

Zeljko Durdevic

Zeljko Durdevic
Senior Editor
EMBO Molecular Medicine

*** Instructions to submit your revised manuscript ***

To submit your manuscript, please follow this link:

<https://embomolmed.msubmit.net/cgi-bin/main.plex>

When preparing your revised manuscript, please refer to our guidelines: <https://link.springer.com/journal/44321/submission-guidelines#cms-Revised-submissions>. We perform an initial quality control of all revised manuscripts before re-review; failure to include requested items will delay the evaluation of your revision.

We require:

- 1) A .docx formatted version of the manuscript text (including legends for main figures, EV figures and tables). Please make sure that the changes are highlighted to be clearly visible.
- 2) Individual production quality figure files as .eps, .tif, .jpg (one file per figure). For guidance, download the 'Figure Guide PDF': <https://media.springernature.com/original/springer-cms/rest/v1/content/27825798/data/v1>.
- 3) A .docx formatted letter INCLUDING the reviewers' reports and your detailed point-by-point responses to their comments. As part of the EMBO Press transparent editorial process, the point-by-point response is part of the Review Process File (RPF), which will be published alongside your paper.
- 4) A complete author checklist, which you can download from our author guidelines. Please insert information in the checklist that is also reflected in the manuscript. The completed author checklist will also be part of the RPF.
- 5) Please note that all corresponding authors are required to supply an ORCID ID for their name upon submission of a revised manuscript.
- 6) It is mandatory to include a 'Data Availability' section after the Materials and Methods. Before submitting your revision, primary datasets produced in this study need to be deposited in an appropriate public database, and the accession numbers and database listed under 'Data Availability'. Please remember to provide a reviewer password if the datasets are not yet public.

In case you have no data that requires deposition in a public database, please state so in this section. Note that the Data Availability Section is restricted to new primary data that are part of this study.
- 7) For data quantification: please specify the name of the statistical test used to generate error bars and P values, the number (n) of independent experiments (specify technical or biological replicates) underlying each data point and the test used to calculate p-values in each figure legend. The figure legends should contain a basic description of n, P and the test applied. Graphs must include a description of the bars and the error bars (s.d., s.e.m.).
- 8) At EMBO Press we ask authors to provide source data for the main manuscript figures. You will receive a separate email with instructions for providing source data with your revised manuscript, including how to upload and organize the files.

9) Our journal encourages inclusion of *data citations in the reference list* to directly cite datasets that were re-used and obtained from public databases. Data citations in the article text are distinct from normal bibliographical citations and should directly link to the database records from which the data can be accessed. In the main text, data citations are formatted as follows: "Data ref: Smith et al, 2001" or "Data ref: NCBI Sequence Read Archive PRJNA342805, 2017". In the Reference list, data citations must be labeled with "[DATASET]". A data reference must provide the database name, accession number/identifiers and a resolvable link to the landing page from which the data can be accessed at the end of the reference.

12) Author contributions: You will be asked to provide CRediT (Contributor Role Taxonomy) terms in the submission system. These replace a narrative author contribution section in the manuscript.

13) A Conflict of Interest statement should be provided in the main text.

14) Every published paper includes a 'Synopsis' to further enhance discoverability. Synopses are displayed on the journal webpage and are freely accessible to all readers. They include a short stand first (maximum of 300 characters, including space) as well as 2-5 one-sentences bullet points that summarizes the paper. Please write the bullet points to summarize the key NEW findings. They should be designed to be complementary to the abstract - i.e. not repeat the same text. We encourage inclusion of key acronyms and quantitative information (maximum of 30 words / bullet point). Please use the passive voice. Please attach these in a separate file or send them by email, we will incorporate them accordingly.

15) Include a Reagents and Tools Table as part of the Methods section, which can be downloaded from our author guidelines.

Graphs 800-1,200 DPI
Photos 400-800 DPI
Colour (only CMYK) 300-400 DPI"

*Additional important information regarding figures and illustrations can be found at
<https://media.springernature.com/original/springer-cms/rest/v1/content/27825798/data/v1>

The authors addressed the remaining editorial issues.

9th Feb 2026

Dear Dr. Nguyen,

We are pleased to inform you that your manuscript is accepted for publication and is now being sent to our publisher to be included in the next available issue of EMBO Molecular Medicine.

You may qualify for financial assistance for your publication charges - either via a Springer Nature fully open access agreement or an EMBO initiative. Check your eligibility: <https://link.springer.com/journal/44321/how-to-publish-with-us>

Zeljko Durdevic
Senior Editor
EMBO Molecular Medicine

>>> Please note that it is EMBO Molecular Medicine policy for the transcript of the editorial process (containing referee reports and your response letter) to be published as an online supplement to each paper. If you do NOT want this, you will need to inform the Editorial Office via email immediately. More information is available here: <https://link.springer.com/partners/embo-press/editorial-policies#Peer%20review>